# Heterojunction formed via 3D-to-2D perovskite conversion for photostable wide-bandgap perovskite solar cells

Jin Wen[1], Yicheng Zhao [2], Pu Wu[1], Yuxuan Liu[1], Xuntian Zheng[1], Renxing Lin[1], Sushu Wan [3], Ke Li[3], Haowen Luo[1], Yuxi Tian [3], Ludong Li [1] & Hairen Tan [1]✉

Light-induced halide segregation constrains the photovoltaic performance and stability of wide-bandgap perovskite solar cells and tandem cells. The implementation of an intermixed two-dimensional/three-dimensional heterostructure via solution post-treatment is a typical strategy to improve the efficiency and stability of perovskite solar cells. However, owing to the composition-dependent sensitivity of surface reconstruction, the conventional solution post-treatment is suboptimal for methylammonium-free and cesium/bromide-enriched wide-bandgap PSCs. To address this, we develop a generic three-dimensional to two-dimensional perovskite conversion approach to realize a preferential growth of wider dimensionality ($n \geq 2$) atop wide-bandgap perovskite layers (1.78 eV). This technique involves depositing a well-defined $MAPbI_3$ thin layer through a vapor-assisted two-step process, followed by its conversion into a two-dimensional structure. Such a two-dimensional/three-dimensional heterostructure enables suppressed light-induced halide segregation, reduced non-radiative interfacial recombination, and facilitated charge extraction. The wide-bandgap perovskite solar cells demonstrate a champion power conversion efficiency of 19.6% and an open-circuit voltage of 1.32 V. By integrating with the thermal-stable $FAPb_{0.5}Sn_{0.5}I_3$ narrow-bandgap perovskites, our all-perovskite tandem solar cells exhibit a stabilized PCE of 28.1% and retain 90% of the initial performance after 855 hours of continuous 1-sun illumination.

All-perovskite tandem solar cells, which consist of a wide-bandgap (WBG, ~1.8 eV) perovskite top cell paired with a narrow-bandgap (NBG, ~1.2 eV) perovskite bottom cell, offer the potential for higher efficiency than the SQ limit of single-junction solar cell while maintaining the benefits of low-cost solution processing[1,2]. In the past few years, the power conversion efficiency (PCE) of all-perovskite tandem solar cells has already surpassed that of single-junction perovskite solar cells (PSCs), positioning themselves as a promising next-generation

photovoltaic (PV) technology[3–8]. Despite the significant advancements in PCE, instability issues still hinder the commercialization of all-perovskite tandem solar cells[9].

The WBG subcells, which alloy a high content of bromide (~40%), suffer from severe light-induced halide segregation under continuous illumination[10–12], thereby limiting the operational lifetime of tandem PSCs. Various strategies have been employed to address this challenge, such as passivating the interface[13,14], relaxing lattice strain[15,16],

[1]National Laboratory of Solid State Microstructures, Frontiers Science Center for Critical Earth Material Cycling, College of Engineering and Applied Sciences, Nanjing University, Nanjing 210023, China. [2]State Key Laboratory of Electronic Thin Films and Integrated Devices, School of Electronic Science and Engineering, University of Electronic Science and Technology of China, Chengdu 610054, China. [3]School of Chemistry and Chemical Engineering, Nanjing University, Nanjing 210023, China. ✉e-mail: hairentan@nju.edu.cn

modifying compositions[17–22], and developing new transport layers[14,23]. Such efforts have, to some extent, mitigated light-induced halide segregation. Nevertheless, understanding the fundamental mechanism of phase segregation in full device structure is still a matter of ongoing investigation.

In single-junction PSCs with bandgaps ~1.5 eV, carrier charges trapped in perovskite films were found to induce irreversible degradation[24–26]. Here, we reveal that charge accumulation accelerates the halide segregation in WBG PSCs under illumination, originating from the mismatch of energy levels and high trap density at contacting heterointerface with the electron transport layer (fullerene, $C_{60}$). This prompts us to design the perovskite/$C_{60}$ interface with facilitated charge extraction and durable interfacial contact.

The construction of an intermixed 2D/3D heterostructure, accomplished by spin-coating long-chain ammonium ligands dissolved in isopropyl alcohol (IPA) onto the surface of a 3D perovskite, has been widely utilized to optimize contact heterointerfaces in PSCs[27–29]. The solution-based post-treatment involves the dissolution of the 3D perovskite, leading to a more $PbI_2$-rich surface suitable for the ammonium ligands to intercalate[30]. However, when applied to the MA-free and Cs/Br-enriched WBG perovskites, recognized for their better photostability and thermal stability[15,20,31], this technique yields inconsistent results due to intricate surface chemistry with increased activation energy for surface reconstruction growth.

Herein, we present a generic 3D-to-2D perovskite conversion approach: depositing a well-defined $MAPbI_3$ thin layer by a hybrid evaporation/solution method, followed by its complete transformation into a 2D structure via a long-chain ammonium ligand. This approach overcomes the composition-dependent surface sensitivity of conventional solution-processing strategies and enables the preferential growth of wider dimensionality ($n \geq 2$) on the MA-free WBG perovskite layer. The resulting 2D/3D heterostructure effectively accelerates the charge extraction at the perovskite/$C_{60}$ interface and thereby suppresses light-induced halide segregation. We achieved a champion PCE of 19.6% and an impressive $V_{oc}$ of 1.32 V in 1.78-eV WBG

PSCs. Meanwhile, the devices exhibited superior operational stability – maintaining 95% of their initial PCE after 1000 h operation at the maximum power point (MPP). These enabled us to realize an impressive PCE of 28.1% in monolithic all-perovskite tandem solar cells.

## Results

### The impact of charge accumulation on halide segregation

To understand the mechanism of light-induced halide segregation in WBG PSCs, we first investigated the photostability at MPP and open-circuit (OC) conditions for 1.55-, 1.68-, and 1.78 eV-bandgap devices (Fig. 1a, b). The p-i-n device structure is glass/ITO/$NiO_x$-SAM/perovskite/$C_{60}$/ALD-$SnO_2$/Cu (SAM: self-assembled monolayer). The photo-generated carriers under OC condition are expected to be unextractable and would accumulate at a contacting selective heterointerface, in contrast to the MPP condition where carriers are extracted from the absorber during operation[32–34]. For devices with different bandgaps, we found that devices under MPP tracking showed significantly less degradation than those under OC condition (Fig. 1c, d and Supplementary Fig. 1). Furthermore, the difference in degradation rate under MPP and OC conditions of WBG PSCs was more pronounced than that of their narrower bandgap counterparts. The X-ray diffraction (XRD) measurement indicated that the WBG device exhibited significant halide segregation after 80 hours of continuous light illumination under the OC condition, while those under the MPP condition only showed a decline in crystallinity (Supplementary Fig. 2).

To explore whether excess charge carrier induced instability, we then investigated the photoluminescence (PL) evolution of $Cs_{0.2}FA_{0.8}Pb(I_{0.6}Br_{0.4})_3$ device under OC and short-circuit (SC) conditions. Under SC conditions, there is a minimal built-in electric field within the devices, thus allowing for rapid carrier extraction. We used a 532 nm laser with an intensity equivalent to 15 suns, in order to accelerate the degradation process. The PL spectra under the OC condition exhibited a growing red shift from 700 nm to over 750 nm, corresponding to the formation of an I-rich phase after 10 min of continuous illumination (Fig. 1e). In contrast, the device illuminated

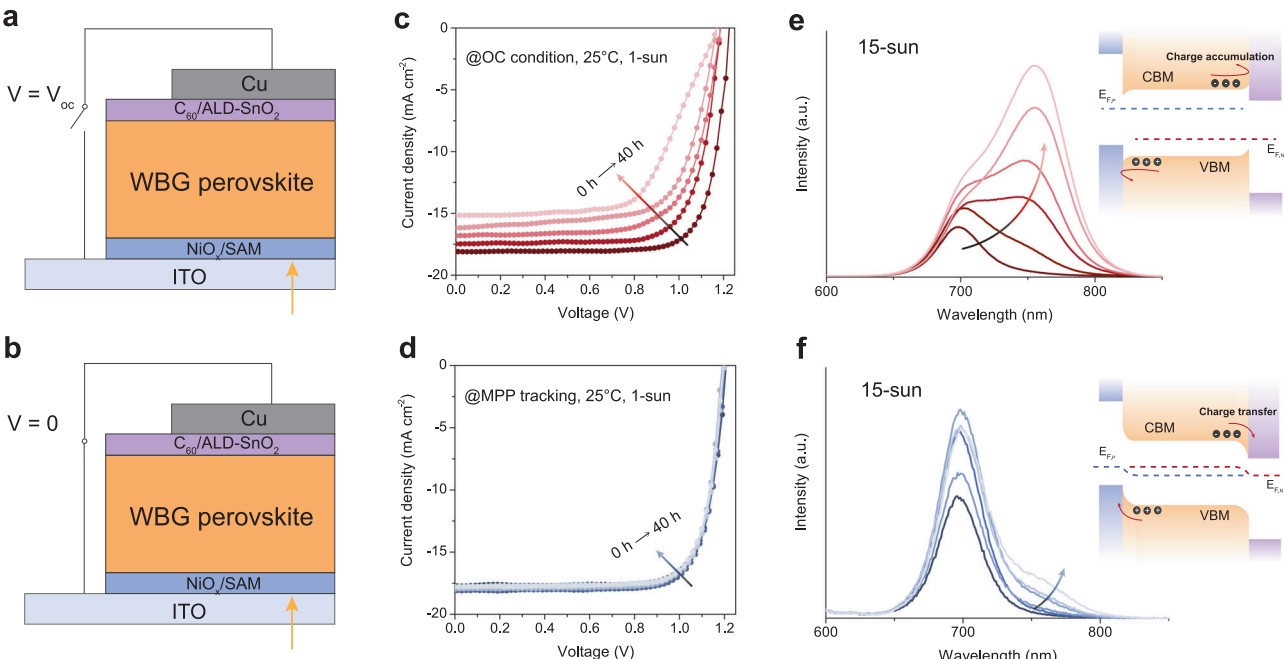

**Fig. 1 | Impact of charge accumulation on light-induced halide segregation in WBG perovskite solar cells. a, b** Device structure of $Cs_{0.2}FA_{0.8}Pb(I_{0.6}Br_{0.4})_3$ PSCs for photostability study under **a** open-circuit and **b** short-circuit conditions. **c, d** J–V curves evolution of WBG device under illumination at **c** open-circuit and **d** MPP tracking conditions. **e, f** PL spectra of perovskite device under illumination at **e** open-circuit and **f** short-circuit conditions. The samples were excited under a 532 nm laser for 10 min. The insets show schematic illustrations of band alignments in devices under illumination at open-circuit and short-circuit conditions. CBM conduction band minimum, VBM valence band maximum, $E_{F,N}$ electron quasi-fermi level, $E_{F,P}$ hole quasi-fermi level.

under SC conditions exhibited a significantly lower extent of halide segregation compared to the device under OC conditions (Fig. 1f). When the laser intensity was set to 1 sun, the difference in WBG device stability between OC and SC condition can still be observed (Supplementary Fig. 3). This result is consistent with the $J$–$V$ curve evolution of the WBG devices under MPP tracking, which showed excellent stability over a period of 40 hours (Fig. 1d). Combining these results, we suggested that in WBG PSCs, charge accumulation had a profound impact on light-induced halide segregation.

We also investigated the photostability of WBG devices having different compositions: $Cs_{0.2}FA_{0.8}Pb(I_{0.6}Br_{0.4})_3$ and $MA_{0.3}FA_{0.7}Pb(I_{0.6}Br_{0.4})_3$ (Supplementary Fig. 4). We fabricated and subjected the devices to light-soaking under the simulated 1-sun illumination at OC conditions in the glove box. Previous studies have demonstrated that CsFA WBG perovskite films exhibited better photostability than their MAFA counterpart[35]. However, we observed only a minor difference in device stability under OC conditions, indicating that halide segregation could be driven by the charge accumulation in complete devices. We also observed that under OC conditions, the PCE loss of WBG PSCs mainly resulted from the reduction of short-circuit current density ($J_{sc}$) and fill factor (FF) rather than the open-circuit voltage ($V_{oc}$) (Fig. 1c and Supplementary Fig. 3). The intense emission peak originating from the enriched I-phase offers a credible explanation for the maintained $V_{oc}$ (Supplementary Fig. 5). In addition, the integrated $J_{sc}$ values from the external quantum efficiency (EQE) curve are significantly higher than those from the $J$–$V$ tests after the same aging time (Supplementary Fig. 4e, f). Time-resolved PL (TRPL) spectra also indicate an increased lifetime of aged devices (Supplementary Fig. 6a). It is worth noting that the increased lifetime could be attributed to both aging effects and improved material quality. To delve deeper, we performed calculations for the differential lifetime (Supplementary Fig. 6b) to distinguish between charge extraction and trap-assisted recombination. The initial interval at shorter times is primarily influenced by the transfer of carriers from the bulk into the transport layer, while the subsequent interval at longer delay times is dominated by interfacial recombination. The result strongly suggests a suppressed carrier extraction towards the electrodes following the aging process – in connection with the significantly decreased $J_{sc}$ and FF. The above results suggest that charge accumulation, caused by insufficient carrier extraction, is a major driving force on the performance degradation of WBG PSCs.

According to previous reports, charge carrier tends to be trapped at the heterointerface of the perovskites and charge-extraction materials[25,36,37]. We then used a 375 nm laser for TRPL study from both sides of the device, where over 90% of photons are absorbed within the first 30 nm of the perovskite film due to the high absorption coefficient at this wavelength. After light soaking, we noticed a rise in PL lifetime when laser light was incident from the HTL side, but there was barely any change in the PL decay when the laser was incident from the ETL side (Supplementary Fig. 7). Given the swift transfer of electrons (holes) generated at the interface towards adjacent transport layers, TRPL reveals the efficiency of holes (electrons) extraction to the opposite transport layer. The results suggest an inadequate transport of photogenerated electrons from the perovskite layer to ETL[38,39]. The $V_{oc}$ loss at the perovskite/$C_{60}$ interface was substantially higher than that at the $NiO_x$-SAM/perovskite interface (Supplementary Fig. 8), indicating a higher trap density at the perovskite/ETL interface. We speculate that the electron extraction is inhibited at the perovskite/$C_{60}$ interface under illumination, which in turn accelerates the halide segregation.

## 2D/3D heterojunction formed via 3D-to-2D conversion

Within the pin device structure, the integration of 2D structures with higher n-values is a promising approach to effectively reduce the bandgap and tune the energy level alignment, thereby enhancing

electron transport at the perovskite/$C_{60}$ interface[13,27,29,40–43]. We initially post-treated the surface defects of the 3D WBG perovskite films by applying PEAI molecules to form Ruddlesden-Popper phase 2D-perovskite layers. However, we found that the $J_{sc}$ of WBG PSCs was decreased and the PCE was not significantly improved after PEAI treatment, indicating an inhibited charge extraction (Supplementary Fig. 9). We then investigated the formation of 2D perovskites using XRD measurements. Supplementary Fig. 10 showed a diffraction peak at 5.4° that belongs to $n = 1$ 2D perovskite, in accordance with previous studies[27,43]. As the concentration of PEAI further increased, a diffraction peak appeared at 4.7° which is ascribed to the unreacted PEAI. The decrease of $J_{sc}$ after PEAI passivation can be attributed to the $n = 1$ 2D perovskite layer, which blocks electron transport in the p-i-n structure due to a wide bandgap of ~2.4 eV and energy level mismatch with $C_{60}$[29,42].

We further explored the formation of 2D structures using PEAI solution spin-coated directly upon varying perovskite compositions, especially WBG compositions (Fig. 2a, b). The PEAI surface treatment primarily forms an $n = 1$ 2D structure on the CsFA-based compositions. As the Cs content increased beyond 35% in the A site, only the peak of unreacted PEAI was observed in the XRD pattern, indicating that 2D perovskite was not able to form on the surface. We found that only the $MAPbI_3$ perovskite film surface produced 2D perovskite structures containing a majority of $n = 3$ RDPs. It is also worth noting that the film morphology changed after IPA washing (Supplementary Fig. 11). We noticed the appearance of $PbI_2$ correlates with the formation of 2D structures, owing to the fact that the formation of 2D structure involves IPA-assisted dissolution and reconstruction of the 3D perovskite surface[30,44]. The XRD pattern confirmed the appearance of $PbI_2$ peaks in the $MAPbI_3$ film after IPA cleaning, while the Cs-Br WBG film remained unchanged (Supplementary Fig. 12). We attribute this observation to the varying solubility of A-site cations in IPA: CsFA-based perovskites exhibit significantly lower solubility compared to their MA-based counterparts (Supplementary Fig. 13). These experiments highlight the significant impact of $PbI_2$ and MAI on the formation of high n-value 2D structures, a phenomenon consistent with findings reported in the literature on 2D perovskites[29,45,46].

We then devised a strategy to directly construct a quasi-2D ($n \geq 2$) perovskite layer on CsFA-based WBG perovskite film via a hybrid evaporation/spin-coating deposition method (Fig. 2c). For an optimized performance of WBG PSCs, a thin layer of $MAPbI_3$ was firstly deposited by a two-step method with evaporated $PbI_2$ (10 nm) and spin-coated MAI (2 mg mL$^{-1}$). Then, 2 mg mL$^{-1}$ PEAI was spin-coated on top to transform the thin $MAPbI_3$ perovskite layer into a 2D layer (see details in Supplementary Note 1, Supplementary Fig. 14–20). Top-view SEM images revealed a substantial surface morphology evolution during the three-step post-treatment (Fig. 2d). The XRD peak of VAQ-2D was detected when we utilized the same processing to deposit 2D layer on the glass, confirming that the formation of 2D perovskite layers with $n \geq 2$ (Supplementary Fig. 20). As the $PbI_2$ layer was thickened to 20 nm, we also observed the appearance of PL emission peaks at 560, 610 and 650 nm (corresponding to $n = 2$, 3, and 4, respectively) in films, providing further confirmation of the VAQ-2D structure. (Fig. 2e). In comparison, PL spectra of standard-2D/3D perovskites showed a dominant emission peak at 500 nm, corresponding to $n = 1$ 2D perovskite.

We further investigated the PL quantum yield (PLQY) of perovskite films with different stacks (Fig. 2f and Supplementary Fig. 21). This allows us to derive the quasi-Fermi level splitting (QFLS) in the HTL/perovskite and HTL/perovskite/ETL p-i-n device stacks. After depositing a $C_{60}$ layer on top of perovskites, both control and standard 2D stacks showed a significant reduction of ~120 meV and ~70 meV in QFLS (compared to the HTL/perovskite case), which can be ascribed to the non-radiative recombination losses at the perovskite/$C_{60}$ junction. In contrast, the VAQ-2D stack exhibited a much smaller reduction (only

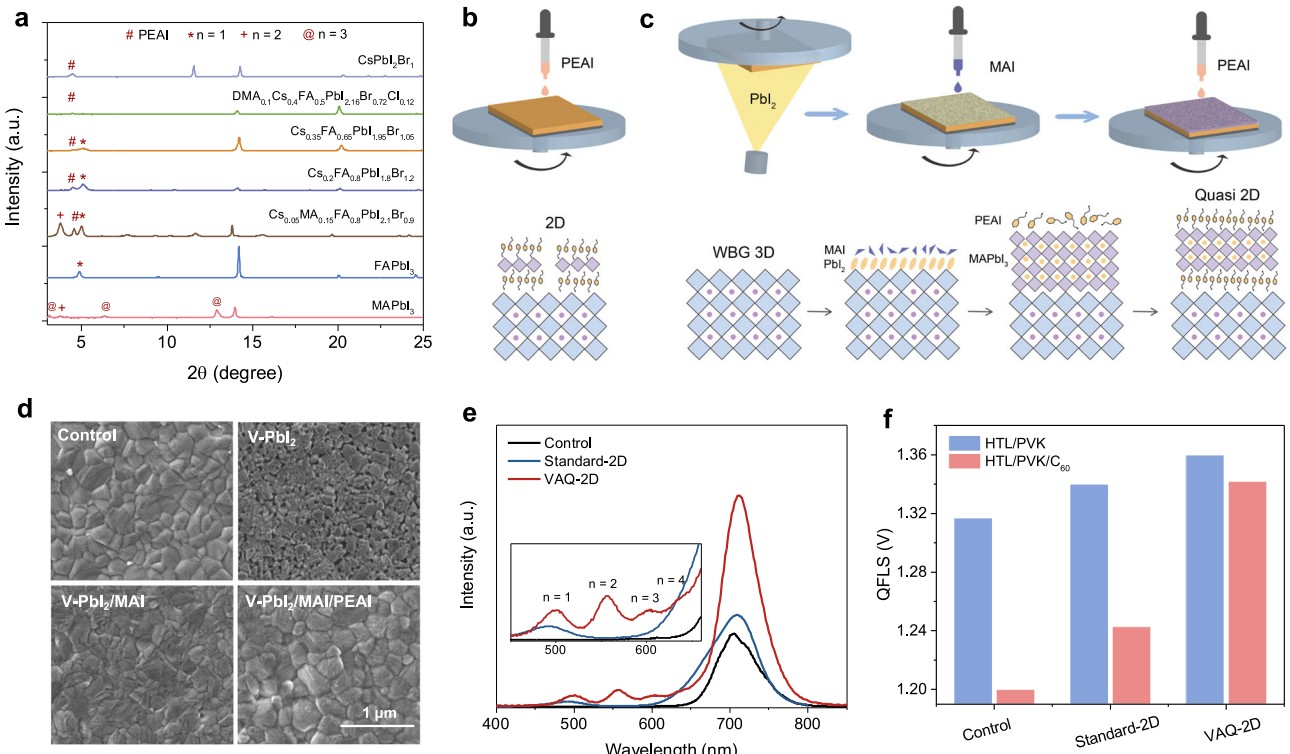

**Fig. 2 | Preparation and characterization of 2D/3D perovskite heterojunction.** **a** XRD patterns of perovskite films with different compositions after solution post-treatment using 5 mg mL$^{-1}$ PEAI. **b**, **c** Schematic of the deposition method of 2D/3D heterojunction: **b** standard solution post-treatment (refer to as standard-2D) and **c** vapor-assisted 3D-to-2D conversion (refer to as VAQ-2D). **d** Corresponding SEM images of WBG perovskite film during the vapor-assisted 3D-to-2D conversion. **e** PL spectra of control, standard-2D, and VAQ-2D films. **f** The calculated QFLS of the HTL/perovskite and HTL/perovskite/ETL junctions.

20 meV) in QFLS, suggesting a substantially suppressed non-radiative recombination at the VAQ-2D perovskite/ETL junction. These results are consistent with the steady-state PL measurements (Supplementary Fig. 22), confirming that the energy loss at the perovskite/$C_{60}$ interface can be substantially mitigated by 2D/3D heterojunction formed via the VAQ-2D strategy.

**Characterization of 2D/3D heterojunction**

We first carried out ultraviolet photoelectron spectroscopy (UPS) and UV-vis measurements to investigate the energy level alignment between WBG perovskite and $C_{60}$ (Fig. 3a and Supplementary Figs. 23, 24). The standard-2D induced a Fermi level downshift and a conduction band minimum (CBM) level upshift, resulting in a large energy barrier for electron transport. In contrast, the VAQ-2D thin film, with Fermi level shifted up and CBM shifted down, exhibited excellent energy level alignment with $C_{60}$ that allows for efficient charge transfer at the electron-selective heterointerface. In addition, the deeper valence band maximum (VBM) of VAQ-2D film can block hole transport and thus reduce electron-hole recombination at the interface (Supplementary Fig. 25).

To further explore the charge extraction at the perovskite/$C_{60}$ interface, we conducted TRPL measurements using an excitation wavelength of 405 nm for both bare films and full device stacks (Fig. 3b, c). The VAQ-2D-treated films exhibited longer lifetimes than standard-2D and non-passivated control samples. We then fabricated HTL/perovskite/$C_{60}$ stacks and investigated the charge extraction by analyzing the fast decay component ($\tau_1$) in PL lifetimes (Fig. 3c). The $\tau_1$ lifetime of the control film was 3.3 ns, which was then increased to 4.0 ns for standard-2D layer, suggesting that $n = 1$ 2D perovskite layer impedes the extraction of electron from perovskite to ETL. In the VAQ-2D stack, $\tau_1$ was reduced to 1.7 ns, implying a considerably improved extraction. In addition, the Shockley–Read–Hall (SRH) lifetime, as

extracted from the slower decay ($\tau_2$) at low excitation fluences, was increased from 14 ns to 31 ns after VAQ-2D treatment, which can be attributed to the suppressed non-radiative recombination. All fitting results of TRPL spectra are summarized in Supplementary Table 1. Based on the aforementioned results, we conclude that the VAQ-2D strategy improved charge extraction and reduced non-radiative recombination at the perovskite/$C_{60}$ interface.

We then carried out steady-state PL measurements with an excitation of 532 nm continuous-wave laser to elucidate the photostability of devices. Figure 3d, e shows the PL peak evolution of standard-2D and VAQ-2D devices under 15-sun illumination for 10 min. The standard-2D device exhibited slightly reduced phase segregation under the OC condition compared to the control device (Fig. 1c). However, upon switching to the SC condition, phase segregation becomes more pronounced than the control device, which could be due to the impeded electron transport. In contrast, the VAQ-2D device showed impressively stable PL spectra under both OC and SC conditions, highlighting its superior photostability.

To further explore photostability, we characterized both bare films and full devices after illumination. PL mapping was first performed for the aged devices with light excitation from the $C_{60}$ side. The VAQ-2D device showed stronger PL emission than those of the control and standard-2D films (Supplementary Fig. 26), indicating a reduced non-radiative recombination. The corresponding XRD patterns of the aged devices were shown in Supplementary Fig. 27, showing severe halide segregation in control and standard-2D devices. However, the XRD pattern of the VAQ-2D device remained only one diffraction peak, though accompanied by peak broadening. Cross-sectional SEM images of the control and standard-2D PSCs after 972-h illumination showed irreversible morphological degradation features, such as cracks and voids, whereas no apparent monography change was observed in the VAQ-2D device (Fig. 3f–h). Such morphological

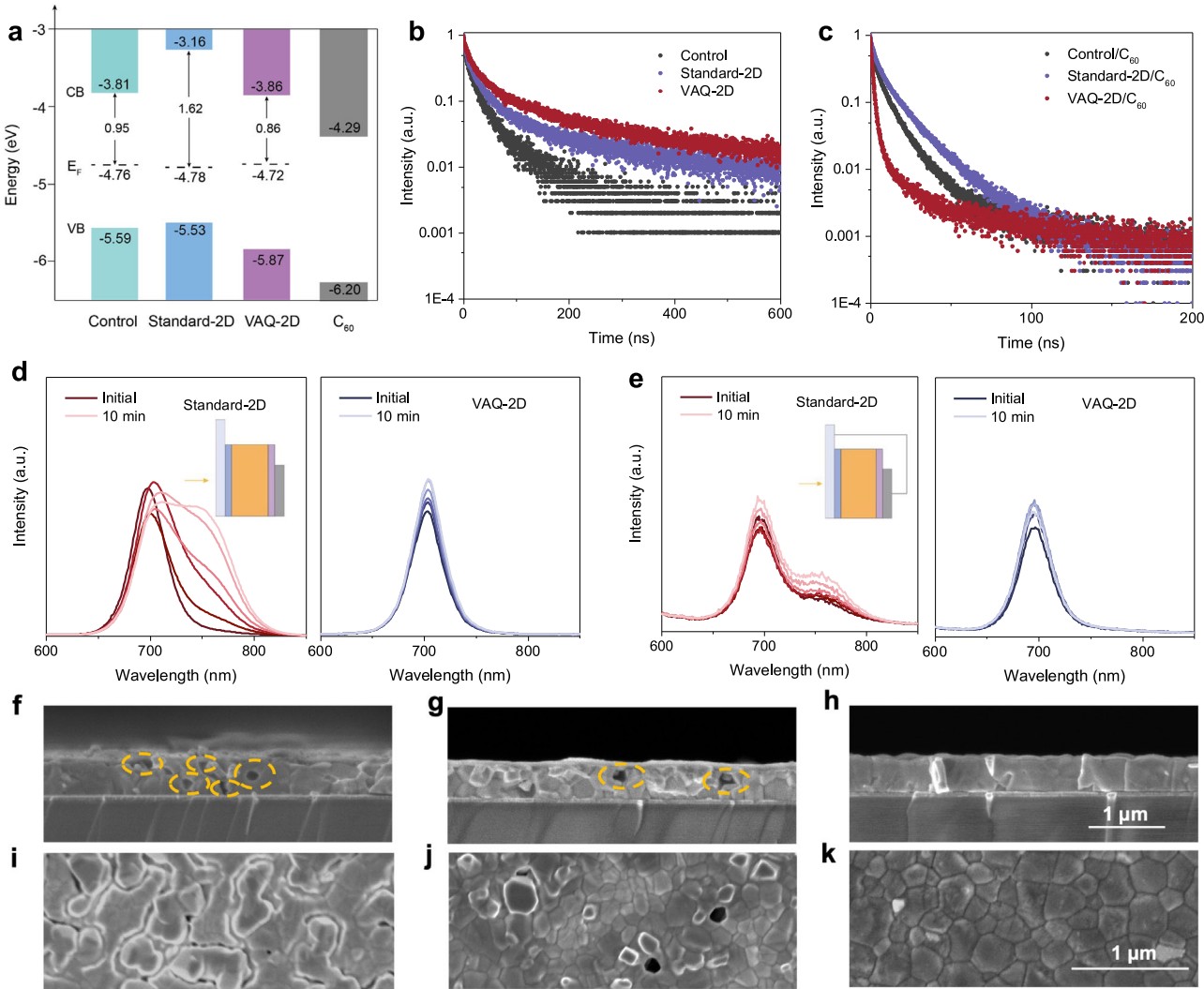

**Fig. 3 | Charge-carrier dynamics and photostability of perovskite heterojunctions. a** Energy level scheme for the control, standard-2D, and VAQ-2D films extracted from UPS data. **b, c** TRPL spectra of (**b**) perovskite films and (**c**) perovskite films coated with $C_{60}$. All films were deposited on glass/NiO$_x$-SAM. **d, e** PL spectral evolution of standard-2D and VAQ-2D device (**d**) under OC, (**e**) under SC condition with a 532 nm laser excitation (intensity equivalent to 15 suns) for 10 min, respectively. **f, g** Cross-sectional SEM images of different devices with ITO/NiO$_x$-SAM/perovskite/$C_{60}$/Cu aged under 1-sun illumination after 972 h: **f** control, **g** standard-2D, and **h** VAQ-2D. Scale bars, 1 μm. **i–k** Top-view SEM images of the (**i**) control, (**j**) standard-2D, and (**k**) VAQ-2D films deposited on HTL and aged under 1-sun illumination after 500 h. Scale bars, 1 μm.

imperfections were not present before light exposure (Supplementary Fig. 28). The crack formation in control films after illumination happened mostly at grain boundaries (Fig. 3i). For standard-2D sample, some clusters appeared at grain boundaries (Fig. 3j). In comparison, degradation features in the VAQ-2D film were negligible (Fig. 3k).

The energy-dispersive X-ray spectroscopy (EDS) mapping (Supplementary Figs. 29–31) shows that the distribution of I and Br in all samples was uniform before soaking. However, after soaking for 500 hours, control and standard-2D films exhibited non-uniform I and Br distributions (Supplementary Fig. 29 and 30). We speculate that the white clusters observed at SEM grain boundaries correspond to regions enriched with iodine after halide segregation in EDS mapping. Remarkably, the VAQ-2D film displayed minimal change in I and Br distribution after the same aging period (Supplementary Fig. 31). X-ray photoelectron spectroscopy (XPS) analysis of Pb 4 f spectra revealed that the binding energies of *Pb 4f5/2* and *Pb 4f7/2* in both the control and standard-2D samples shifted towards lower binding energies by approximately 0.4 eV (Supplementary Fig. 32). This shift is attributed to a weakened binding between Pb and halide ions, likely due to the increased lattice volume resulting from halide segregation. However,

in the case of the VAQ-2D film, the binding energy of Pb exhibited a minor change, approximately 0.05 eV, consistent with the EDS analysis, further emphasizing its enhanced photostability.

## PV performance and photostability of solar cells

We fabricated WBG PSCs with an inverted device structure of ITO/NiO$_x$-SAM/perovskite/$C_{60}$/BCP/Cu. Fig. 4a and Supplementary Note 2 compare the PV parameters of solar cells (15 devices for each type) with different 2D passivation fabricated over several identical runs. Compared to control devices, the standard-2D treatment improved $V_{oc}$ and FF but decreased $J_{sc}$, which was mainly due to the impeded charge extraction. VAQ-2D treatment resulted in the best $V_{oc}$, $J_{sc}$, FF, and thus PCE, which is consistent with our finding that quasi-2D perovskite structures result in fewer trap densities and more efficient charge extraction. Figure 4b shows the PCE histogram of 30 VAQ-2D devices processed among several batches, indicating an average PCE of 19.1% and good reproducibility. The best VAQ-2D device showed a PCE of 19.6% (stabilized 19.6%, Supplementary Fig. 35), with a $V_{oc}$ of 1.324 V, a $J_{sc}$ of 17.9 mA cm$^{-2}$, and an FF of 83.0% under reverse scan. The ideality factor n of 1.17 for the VAQ-2D device is smaller than that of 1.73 for the

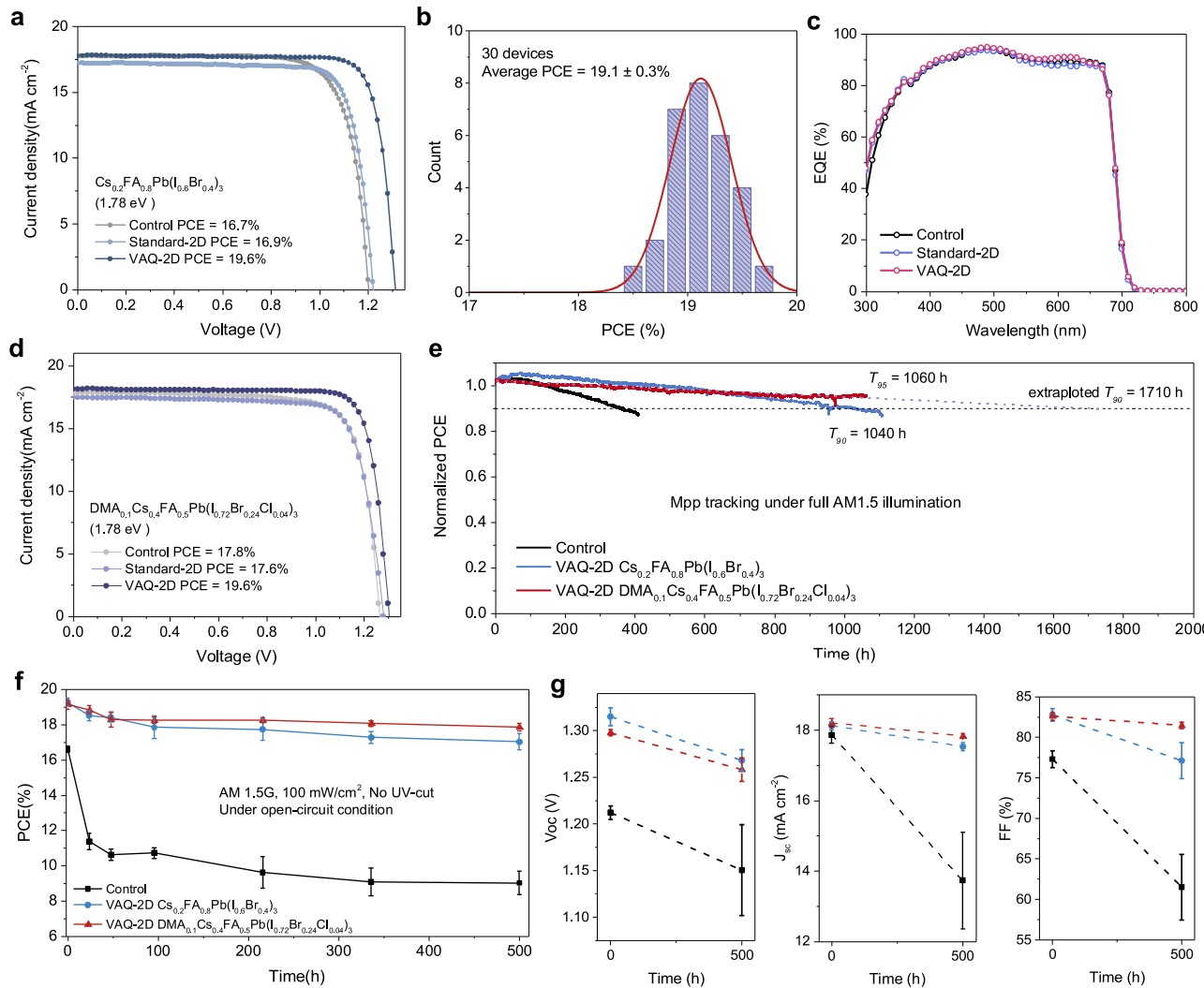

**Fig. 4 | PV performance and photostability of WBG PSCs. a** $J$–$V$ curves from control, standard-2D, and VAQ-2D-treated $Cs_{0.2}FA_{0.8}Pb(I_{0.6}Br_{0.4})_3$ devices. **b** PCE distribution of VAQ-2D-treated devices. **c** EQE curves of WBG devices with standard-2D or quasi-2D passivation. $J$–$V$ curves of the champion VAQ-2D device with different bandgaps. **d** $J$–$V$ curves from control, standard-2D and VAQ-2D-treated $DMA_{0.1}Cs_{0.4}FA_{0.5}Pb(I_{0.72}Br_{0.24}Cl_{0.04})_3$ devices. **e** MPP tracking was measured with the encapsulated control and VAQ-2D devices under full solar illumination (AM 1.5 G, 100 mW cm$^{-2}$) in ambient conditions. **f, g** PCE and detailed parameters evolution of the encapsulated control and VAQ-2D devices measured at open-circuit conditions under full solar illumination (AM 1.5 G, 100 mW cm$^{-2}$) in ambient air. The error bars represent the standard deviation of the PCE measured from five devices.

reference device, suggesting reduced trap-assisted recombination in the device with VAQ-2D passivation (Supplementary Fig. 36). Fig. 4c presents the EQE spectra of the control, standard-2D and VAQ-2D devices, with the integrated $J_{sc}$ value of 17.7, 17.5, and 18.0 mA cm$^{-2}$, respectively, in good agreement with $J$–$V$ characterization.

We further demonstrated that our proposed passivation approach was universal for various perovskite compositions. As shown in Fig. 4d, for the photostable $DMA_{0.1}Cs_{0.4}FA_{0.5}Pb(I_{0.72}Br_{0.24}Cl_{0.04})_3$ composition in our previous work[15], the VAQ-2D strategy significantly improves the efficiency while the standard-2D treatment has no optimal effect. The champion VAQ-2D device for $DMA_{0.1}Cs_{0.4}FA_{0.5}Pb(I_{0.72}Br_{0.24}Cl_{0.04})_3$ achieved a PCE of 19.6%, with a $V_{oc}$ of 1.302 V, a $J_{sc}$ of 18.1 mA cm$^{-2}$ and a high FF of 83.3% under reverse scan. We also achieved a champion PCE of 24.6% for a $Cs_{0.05}FA_{0.95}PbI_3$ device (bandgap of 1.51 eV) in the p-i-n structure, while the PCE of control and the standard-2D device is 22.5% and 22.2%, respectively (Supplementary Fig. 37). Our approach provides new perspectives on how to construct the VAQ-2D layer on surfaces of low-solubility compositions.

Device stability is crucial for the commercialization of PSCs. We then evaluated the stability of encapsulated devices under continuous illumination without an ultraviolet filter (Fig. 4e). All encapsulated devices were aged in an ambient atmosphere at about 25 °C. The VAQ-2D treated $Cs_{0.2}FA_{0.8}Pb(I_{0.6}Br_{0.4})_3$ device retained 90% of its performance after about 1020 h of light illumination, while the control device exhibited a $T_{90}$ of 385 h. For $DMA_{0.1}Cs_{0.4}FA_{0.5}Pb(I_{0.72}Br_{0.24}Cl_{0.04})_3$, after 1060 hours of continuous operation under 1-sun illumination, the VAQ-2D-treated device still retained 95% of its initial PCE, with a linear extrapolation to a $T_{90} > 1700$ hours. The stability of standard-2D devices, as shown in Supplementary Fig. 38, deteriorated more rapidly compared to the control devices, and we attribute this to the degradation caused by interface electron blocking. We further found that devices were generally less stable under OC conditions than those under MPP tracking, due to the effect of the charge accumulation. The VAQ-2D devices with $Cs_{0.2}FA_{0.8}Pb(I_{0.6}Br_{0.4})_3$ and $DMA_{0.1}Cs_{0.4}FA_{0.5}Pb(I_{0.72}Br_{0.24}Cl_{0.04})_3$ maintained 88% and 93% of their initial PCE after 500 hours of MPP tracking under continuous light illumination compared to the control devices, which degraded roughly

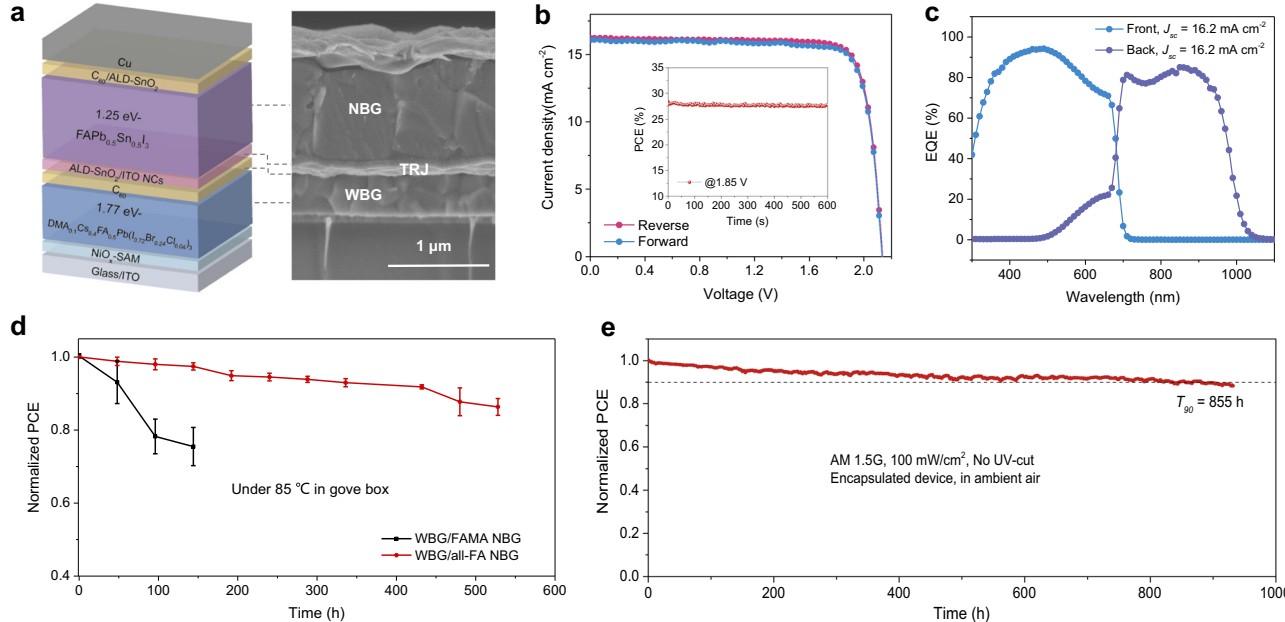

**Fig. 5 | PV performance and stability of all-perovskite tandem solar cells.**
**a** Schematic of the solar cell structure and the corresponding cross-sectional SEM image of an all-perovskite tandem solar cell. **b** *J*–*V* curves of the best-performing tandem solar cell, showing PCEs of 28.1% and 28.1% under reverse and forward scans, respectively. The inset shows the power output of the champion tandem device for 600 s, exhibiting a stabilized PCE of 28.1%. **c** EQE curves of the best-performing tandem cell, showing integrated $J_{sc}$ values of 16.2 and 16.2 mA cm$^{-2}$ for the front and back subcells, respectively. **d** Variation of PCEs for encapsulated tandem devices with FAMA and all-FA NBG subcell under 85 °C stress. **e** MPP tracking of an encapsulated tandem solar cell for 930 h in ambient air under full simulated AM1.5 solar illumination (100 mW cm$^{-2}$) without an ultraviolet filter.

to 54% of their initial PCE over the same timespan (Fig. 4f). Moreover, the VAQ-2D treatment inhibited the decrease of $J_{sc}$ and FF under the OC condition, indicating the suppressed degradation and more effective charge extraction (Fig. 4g).

We fabricated monolithic all-perovskite tandem solar cells using the optimized WBG perovskite layers (Fig. 5a, see details in the Methods). We chose NBG perovskites with a composition of FAPb$_{0.5}$Sn$_{0.5}$I$_3$, considering their much-improved thermal stability[47] and an absorption edge at ~1.25 eV. The NBG solar cells exhibited a PCE of 21.3%, with a $V_{oc}$ of 0.843 V, a $J_{sc}$ of 32.0 mA cm$^{-2}$ and an FF of 79.3% (Supplementary Fig. 39). The configuration of the tandem is glass/ITO/NiO$_x$-SAM/WBG perovskite/C$_{60}$/ALD-SnO$_2$/ITO NCs/NBG perovskite/C$_{60}$/ALD-SnO$_2$/Cu (indium tin oxide nanocrystals abbreviated as ITO NCs). We fabricated 30 tandem devices (area ~0.049 cm$^2$), showing an average PCE of 27.7 ± 0.3% and a relatively narrow PCE distribution (Supplementary Fig. 40). Figure 5b, c presents the *J*–*V* and EQE curves of the best-performing tandem solar cell, and the corresponding PV parameters are summarized in Table 1. The champion tandem device exhibited a high PCE of 28.1% under reverse scan (with a $V_{oc}$ of 2.135 V, a $J_{sc}$ of 16.2 mA cm$^{-2}$, and an FF of 81.5%), showing a stabilized PCE of 28.1%. The integrated $J_{sc}$ values of the front and back subcells from EQE curves are 16.2 and 16.2 mA cm$^{-2}$, respectively, agreeing well with the $J_{sc}$ determined from *J*–*V* measurements. We also achieved a champion

PCE of 27.3% for large-area tandem (1.05 cm$^2$ aperture area), with an average PCE of 26.7 ± 0.3% (Supplementary Fig. 41).

We evaluated the thermal stability under 85 °C of our encapsulated devices in the glove box. Figure 5d demonstrates the superior stability of tandem with all-FA NBG subcell under thermal stressing, exhibiting 87% of its initial PCE after 528 h. Whereas the device with FAMA NBG subcell degrades to 75% of its initial PCE after 144 h, due to the volatile nature of MA and an unfavorable reaction between PEDOT:PSS and perovskite[48,49]. To assess the photostability of tandem solar cells, we tested operating stability using MPP tracking of an encapsulated tandem in ambient air (Fig. 5e). Under AM1.5 G 1 sun illumination at the MPP, the device retained 90% of its initial PCE after 855 hours of continuous operation. During MPP operation, the device's temperature is maintained at ~31 °C with the assistance of the fan cooling (Supplementary Fig. 42). This promising operating stability (longest $T_{90}$ recorded to date) combined with high PCE represents a crucial step toward the practical application of all-perovskite tandem solar cells (Supplementary Fig. 43).

## Discussion
In this work, we reveal that halide segregation in WBG PSCs is correlated to charge accumulation and that accelerating charge extraction improves photostability. We developed a 3D-to-2D perovskite conversion approach to tailor the dimensionality (*n*) of 2D fragments in 2D/3D heterostructure, which effectively accelerated the charge extraction and reduced nonradiative recombination at the perovskite/C$_{60}$ heterointerface. The optimized WBG single-junction cells, with a high PCE of 19.6% and an impressive $V_{oc}$ of 1.32 V, exhibited promising operational stability, maintaining 95% of their performance after 1100 h of MPP operation under full 1 sun illumination. The improved WBG perovskites further enabled the fabrication of monolithic all-perovskite tandem solar cells with a stabilized PCE of 28.1% and a high $V_{oc}$ of 2.135 V, retaining >90% of their initial efficiencies after 855 hours of operation at the MPP. This study notably reveals the degradation mechanism of WBG perovskite devices and proposes an

## Table 1 | PV performance of champion all-perovskite tandem solar cell and corresponding single-junction subcells

|  | Scan direction | $V_{oc}$ (V) | $J_{sc}$ (mA cm$^{-2}$) | FF (%) | PCE (%) |
|---|---|---|---|---|---|
| WBG subcell | Reverse | 1.324 | 17.9 | 83.0 | 19.6 |
|  | Forward | 1.317 | 17.9 | 82.6 | 19.5 |
| NBG subcell | Reverse | 0.843 | 32.0 | 79.3 | 21.3 |
|  | Forward | 0.841 | 31.9 | 78.7 | 21.1 |
| 2 T tandem | Reverse | 2.135 | 16.2 | 81.5 | 28.1 |
|  | Forward | 2.132 | 16.1 | 82.0 | 28.1 |

unprecedented technical approach to simultaneously improving efficiency and stability. The strategy offers a crucial step toward the full commercialization of all PSCs.

## Methods

### Materials

All materials were used as received without further purification. $PbI_2$ (99.99%), $PbBr_2$ (99.99%), $PbCl_2$ (99.99%), 4PACz (>98.0%) and MeO-2PACz (>98.0%) were purchased from TCI Chemicals. $SnI_2$ (99.999%) was purchased from Alfa Aesar. $SnF_2$ (99.9%), formamidine sulfinic acid (FSA, ≥98%), CsI (99.999%), DMF, (99.8% anhydrous), DMSO (99.9% anhydrous), IPA (99.8% anhydrous), ethyl acetate (99.8% anhydrous) and ITO NCs (<100 nm particle size (DLS) were purchased from Sigma-Aldrich. The organic halide salts (FAI, MAI, PEAI, DMAI, MACl) were purchased from GreatCell Solar Materials (Australia). Guanidinium thiocyanate (99%) and BCP (>99% sublimed) were purchased from Xi'an Polymer Light Technology Corp. PEDOT:PSS aqueous solution (Al 4083) was purchased from Heraeus Clevios (Germany). The $C_{60}$ was purchased from Nano-C. NiO nanocrystals were synthesized according to previous reports.

### Perovskite precursor solution

WBG $Cs_{0.2}FA_{0.8}Pb(I_{0.6}Br_{0.4})_3$ and $DMA_{0.1}Cs_{0.4}FA_{0.5}Pb(I_{0.72}Br_{0.24}Cl_{0.04})_3$ perovskite: the precursor solution (1.2 M) was prepared from four precursors dissolved in mixed solvents of DMF and DMSO with a volume ratio of 4:1. $Cs_{0.2}FA_{0.8}Pb(I_{0.6}Br_{0.4})_3$: The molar ratios for $CsI/FAI/PbI_2/PbBr_2$ were 0.8:0.2:0.4:0.6, respectively. $DMA_{0.1}Cs_{0.4}FA_{0.5}Pb(I_{0.72}Br_{0.24}Cl_{0.04})_3$: The molar ratios for $DMAI/CsI/FAI/PbI_2/PbBr_2$ was 0.1:0.4:0.5:0.625:0.375, respectively. 5 mol% $MAPbCl_3$ (relative to Pb) was additionally added. The precursor solution was stirred at 50 °C for 2 h and then filtered through 0.22 μm PTFE membrane before use.

NBG $FAPb_{0.5}Sn_{0.5}I_3$ perovskite: the precursor solution (2.0 M) was prepared in mixed solvents of DMF and DMSO with a volume ratio of 2:1. The perovskite precursor solution was prepared in an $N_2$-filled glovebox ($H_2O$, $O_2$ < 0.1 ppm). The $FAPb_{0.5}Sn_{0.5}I_3$ perovskite precursor solution was prepared by dissolved FAI, $PbI_2$, $SnI_2$, $SnF_2$, and GuaSCN at a molar ratio of 1:0.5:0.5:0.05: 0.015 in a mixed solvent of DMF and DMSO with the volume ratio of 2:1. The precursor solution was stirred at room temperature for over 2 hours and then filtered through a 0.20-μm PTFE filter before use.

### WBG PSC fabrication

The pre-patterned ITO substrates were cleaned in an ultrasonic cleaner with acetone and isopropanol (IPA) for 15 min, respectively. Then, the ITO flexible substrates were dried by nitrogen flow and fixed onto the glass substrates by double-sided adhesive at four edges to facilitate the fabrication of flexible devices. Before the fabrication of devices, the ITO flexible substrates were treated with UV-ozone for 15 min. NiO nanocrystals were synthesized according to the previous report. NiO nanocrystal (15 mg $ml^{-1}$ in deionized water) solution was spin-coated on cleaned ITO flexible substrates at 4,000 r.p.m. for 20 s, followed by annealing at 100 °C for 10 min in air. Subsequently, the solutions of 2PACz and MeO-2PACz with the same concentration (1 mmol $L^{-1}$ in IPA) were mixed with 3:1 volume ratios and then were spin-coated on the NiO film at 4,000 r.p.m. for 20 s, followed by annealing at 100 °C for 5 min in air. Next, the substrates were transferred into a nitrogen-filled glove box and the WBG perovskite films were deposited with a two-step spin-coating procedure. The first step was 2000 r.p.m. for 10 s with an acceleration of 200 r.p.m. $s^{-1}$. The second step was 6,000 r.p.m. for 40 s with an acceleration of 2000 r.p.m. $s{-1}$. For $Cs_{0.2}FA_{0.8}Pb(I_{0.6}Br_{0.4})_3$ films, chlorobenzene (200 μl) was dropped on the spinning substrate during the second spin-coating step at 20 s before the end of the procedure, followed by annealing at 100 °C for

15 min. For $DMA_{0.1}Cs_{0.4}FA_{0.5}Pb(I_{0.72}Br_{0.24}Cl_{0.04})_3$ films, no antisolvent was used; instead, a gas quenching method was introduced by using a blow dryer H300 made by Xiaomi. The diameter of the air outlet is 9.5 cm. The temperature of hot air is 57 °C and the air flow rate is 20 m/s. The blow dryer was fixed 15 cm above the sample and hot $N_2$ gas was blown on the wet film surface to extract excess DMSO and DMF solvents at 20 s before the end of the second spin-coating procedure. After cooling to room temperature, the substrates were transferred to the evaporation system. $PbI_2$ (10 nm) was sequentially deposited by thermal evaporation (Beijing Technol Science Co.) at the rates of 0.1 Å $s^{-1}$. The thickness of $PbI_2$ layer was read from the evaporation equipment (thickness was calibrated by the evaporation of films on a flat ITO glass substrate). Then, the MAI solution in IPA (2 mg $mL^{-1}$) was drop-casted on perovskite film at 5000 rpm for 20 s, followed by drying at 100 °C for 3 min. Then the cooled films were deposited by PEAI (2 mg/ml) IPA solution at 5000 rpm for 20 s, followed by drying at 100 °C for 3 min. Finally, $C_{60}$ (20 nm), BCP (7 nm), and Cu (150 nm) were sequentially deposited by thermal evaporation (Beijing Technol Science Co.) at the rates of 0.2 Å $s^{-1}$, 0.2 Å $s^{-1}$, and 1.0 Å $s^{-1}$, respectively.

### Monolithic all-perovskite tandem solar cell fabrication

The WBG films were prepared as mentioned above. The substrates were transferred to the evaporation system after the fabrication of all-inorganic perovskite films, and 20-nm-thick $C_{60}$ film was deposited on top by thermal evaporation at a rate of 0.2 Å/s. The substrates were then transferred to the atomic layer deposition (ALD) system (Veeco Savannah S200) to deposit 20 nm $SnO_2$ at 75 °C using precursors of tetrakis(dimethylamino) tin (IV) (99.9999%, Nanjing Ai Mou Yuan Scientific Equipment Co., Ltd) and deionized water. Then 1 wt.% ITO NCs solution (dispersed in isopropanol) was spin-coated on the substrates at 5000 rpm for 15 s and annealed at 100 °C for 5 min in ambient air. Then we immediately transferred the substrates to an $N_2$-filled glovebox for the deposition of $FAPb_{0.5}Sn_{0.5}I_3$ NBG perovskite films with identical procedures used for the single-junction devices. Finally, 20-nm C60, ~15-nm ALD-$SnO_2$, and 150-nm Cu films were sequentially deposited on the surface of NBG perovskite. Details on the deposition of ALD-$SnO_2$ layers can be found in our previous work[50].

### Characterization of solar cells

For single-junction solar cells, the current density-voltage ($J$–$V$) characteristics were measured using a Keithley 2400 sourcemeter under the illumination of the solar simulator (EnliTech, Class AAA) at the light intensity of 100 mW $cm^{-2}$ as checked with NREL calibrated reference solar cells (KG-5 and KG-0 reference cells were used for the measurements of WBG and NBG solar cells, respectively). Unless otherwise stated, the $J$–$V$ curves were all measured in a nitrogen-filled glovebox with a scanning rate of 200 mV $s^{-1}$ (voltage steps of 20 mV and a delay time of 100 ms). The active area was determined by the aperture shade mask (0.049 $cm^2$) placed in front of the solar cells. EQE measurements were performed in ambient air using a QE system (EnliTech) with monochromatic light focused on device pixels and a chopper frequency of 20 Hz. For tandem solar cells, the $J$–$V$ characteristics were carried out under the illumination of a two-lamp high spectral match solar simulator (SAN-EI ELECTRIC, XHS-50S1). The spectrum from the simulator was finely tuned to ensure that spectral mismatch is within 100 ± 3% for each 50-nm interval between wavelength range 400–1000 nm. The solar simulator was set at the light intensity of 100 mW $cm^{-2}$ as checked with a calibrated crystalline silicon reference solar cell with a quartz window (KG-0). EQE measurements were performed in ambient air, and the bias illumination from highly bright LEDs with emission peaks of 850 and 460 nm was used for the measurements of the front and back subcells, respectively. No bias voltage was applied during the EQE measurements of tandems.

## Operational stability tests of solar cells

The operating stability tests were carried out under simulated AM 1.5 G illumination (Class AAA, multi-color LED solar simulator, Guangzhou Crysco Equipment Co. Ltd) with an intensity of 100 mW cm$^{-2}$ using a home-build LabVIEW-based MPP tracking system and a Perturb and observe method in ambient conditions (humidity of 30–50%). The solar cells were encapsulated with a cover glass and UV epoxy (Three Bond, Japan) which was cured under a UV-LED lamp (peak emission at 365 nm) for 3 min. No UV filter was applied during operation. The illumination intensity was regularly calibrated to check the degradation of the LED lamp.

## Other characterizations

XRD patterns were acquired using a Bruker D8 AVANTAGE Diffractometer with Cu K$\alpha$ ($\lambda = 1.54$ Å) radiation. SEM images were obtained using a TESCAN microscope with an accelerating voltage of 2 kV.

PL and TRPL measurements were conducted on a home-built wide-field microscope based on Olymplus IX73. A 532 nm CW diode laser was used as the excitation light for the sample measurements with the laser power density on the film surface of 100 mW cm$^{-2}$. The fluorescence of the samples was collected by a dry objective lens (Olympus LUCPlanFI ×40, NA = 0.6) and detected by an EMCCD camera (iXon Ultra 888, Andor) after passing through 550 nm (ET550LP, Chroma), long-pass filters. We used a pulsed supercontinuum white laser (WL-SC400, Fianium, Southampton, UK) with AOTFs to obtain a monochromatic light of 532 nm (0.5 MHz, 6 nj cm$^{-2}$). A fast avalanche photodiode (APD, MicroPhoton Devices, Bolzano, Italy) coupled with a time-correlated single photon counting module (PicoHarp 300, PicoQuant, Berlin, Germany) was used for the time-resolved PL decay measurements. The PL decay curves were fitted with biexponential components to obtain a fast and a slow decay lifetime.

UPS was performed in an ultrahigh vacuum (UHV) surface analysis system equipped with a fast entry load-lock, a transfer chamber, and an analysis chamber (base pressure ~10$^{-10}$ mbar). UPS employed the He I 21.22 eV as the excitation source with an energy resolution of 50 meV. The work function was derived from the secondary electron cutoff and the ionization potential from the frontier edge of the occupied density states to the vacuum level. Transient photovoltage decays were measured on a homemade system. A 540 nm green light-emitting diode was used to modulate the Voc with a constant light bias, and the repetition rate was set to 2000 Hz. The voltage-decay lifetime was fitted by a single exponential decay. Measurements were carried out in a glovebox using a Keithley 2400 source meter. The LED device characterizations were carried out with a Keithley 2400 source meter and a 100 mm integrating sphere coupled with a spectrometer (Ocean Optics, Spectrum TEQ-EQY).

## Reporting summary

Further information on research design is available in the Nature Portfolio Reporting Summary linked to this article.

# Data availability

The main data generated in this study are provided in the Supplementary Information/Source Data file. All other data supporting the findings of this study are available from the corresponding authors on request. Source data are provided with this paper.

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

## Acknowledgements

H.T. acknowledges the National Key R&D Program of China (2022YFB4200304), the National Natural Science Foundation of China (T2325016, U21A2076, 61974063), Natural Science Foundation of Jiangsu Province (BE2022021, BE2022026, BK20202008, BK20190315). J.W., P.W., and Y.L. thank the Fundamental Research Funds for the Central Universities (0213/14380206; 0205/14380252). H.L. and X.Z. thank the Frontiers Science Center for Critical Earth Material Cycling Fund (DLTD2109), Program for Innovative Talents and Entrepreneurs in Jiangsu. L.L. acknowledges the Guangdong Major Project of Basic and Applied Basic Research (2021B0301030002) and the National Key R&D Program of China (2021YFB3200303). Y.Z. acknowledges the Natural Science Foundation of China (52372177). Y.T., K.L., and S.W. acknowledge the National Natural Science Foundation of China (22073046 and 62011530133).

## Author contributions

H.T. directed the overall project. J.W. conceived the idea fabricated WBG PSCs and conducted the characterization. Y.Z. contributed to the design of experiments. P.W. and R.L. helped with the fabrication of NBG subcells. K.L., S.W., H.L., X.Z., Y.L., Y.T., and L.L. helped with the material characterization. J.W., Y.Z., and H.T. wrote the draft manuscript. All authors read and commented on the manuscript.

## Competing interests

H.T. is the founder, Chief Scientific Officer, and Chairman of Renshine Solar Co., Ltd., a company that is commercializing perovskite PVs. The remaining authors declare no competing interests.
