## [Peer Review File · Nature Communications]

Heterojunction formed via 3D-to-2D perovskite conversion for photostable wide-bandgap perovskite solar cellsREVIEWER COMMENTS

Reviewer #1 (Remarks to the Author):

In this research article, Wen et al. report a study of a heterojunction formed on 1.78-eV wide-bandgap perovskite via vapor-assisted 3D-to-2D perovskite conversion. This conversion involves depositing a well-defined methylammonium lead iodide (MAPbI₃) thin layer through a vapor-assisted two-step process and converting it into a 2D structure by reacting with phenethylammonium iodide (PEAI). This 2D/3D perovskite heterostructured interface suppresses light-induced halide segregation, reduces non-radiative interfacial recombination, and facilitates charge extraction, enabling efficient and stable single-junction wide-bandgap perovskite solar cells with a power conversion efficiency (PCE) of 19.6% and a high open-circuit voltage of 1.32 V. Using the photostable wide-bandgap perovskite subcell, the authors demonstrate all-perovskite tandem solar cells with an impressive stabilized PCE of 28.1%. The tandem cell retains 90% of its initial efficiency after 855 hours of continuous operation under 1-sun illumination.

This work is overall a well-organized study with a relatively novel method to introduce a 2D/3D heterojunction interface to stabilize wide-bandgap perovskites, despite the similarity in the approach to a recent publication from the same group (<https://www.nature.com/articles/s41586-023-06278-z>) and another work on all-inorganic perovskite (<https://www.nature.com/articles/s41560-022-01154-y>). Nonetheless, the work provides somewhat new insight into suppressing phase segregation in wide-bandgap perovskite and demonstrates stable and efficient all-perovskite tandem solar cells. The manuscript could be considered for publication if some technical issues can be adequately addressed.

[1] The section on photostability investigation of wide-bandgap perovskites needs to be clarified. One major concern is the lack of correlation and consistency between the 40 hours of 1-sun device light soaking experiment (Fig. 1c & d) and PL under 15-sun excitation for 10 min (Fig. 1e & f). The authors attempted to attribute the decreased J_{sc} under the V_{oc} -aging condition (Fig. 1c) to light-induced phase segregation (Fig. 1e). However, the validation and interpretation of this hypothesis have two critical issues.

First, the aging conditions (1-sun for 40 h vs. 15-sun for 10 min) in Fig. 1 differ in power density and accumulated light fluence. The high power intensity of 15-sun is known to cause more severe degradation than 1-sun. This aging condition inconsistency also exists in supplementary experiments, i.e., 80, 500, and ~100 h for Supplementary Figures 1, 2, and 3, respectively. The design of the experiment needs better consistency in the aging time and conditions. Moreover, Fig. 1 indicates that degraded PV performance, particularly J_{sc} , is correlated with halide segregation, as evidenced in the PL peak shift. However, Supplementary Figure 4 shows no obvious changes in the PL spectrum of the cell after light soaking, even though the J_{sc} is reduced. Even if the phase segregation occurs and narrower bandgap I-rich phase forms, should the EQE cutoff shift toward a longer wavelength corresponding to the I-rich phase? This is inconsistent with the hypothesis and needs a better explanation.

[2] The statement "After light soaking, we noticed a rise in PL lifetime when laser light was incident from the HTL side, but there was barely any change in the PL decay when the laser was incident from the ETL side (Supplementary Fig. 6). The results suggest an inadequate transport of photogenerated electrons from the perovskite layer to ETL." needs a further explanation.

The data seems to indicate that light soaking does not significantly affect the perovskite/ETL interface (similar PL lifetimes before and after aging) but inhibits hole extraction at the perovskite/HTL interface (longer PL lifetime after aging), which conflicts with the claim.

[3] There are also some inconsistencies in the experimental design for the 2D/3D heterojunction section. The concentration of PEAI (1, 2, 5, 10 mg/ml) and evaporated PbI₂ thickness (10 and 20 nm) vary in different characterizations and are confusing. The samples for XRD measurement (Fig. 2a) used 5 mg/ml PEAI with unknown PbI₂ thickness. Later in the text, a condition of 10 nm PbI₂ and 2 mg/mL PEAI was used for synthesizing a 2D layer. The PL measurement then used 20 nm PbI₂ with an unknown PEAI concentration. Is the 20-nm PbI₂ film with 2D perovskite features shown in Fig. 2e used for the champion solar cell? Could the authors specify how they determine the best PbI₂/PEAI combination for their device optimization and provide relevant characterization evidence?

[4] It is interesting that the standard-2D layer induces a Fermi level downshift and CBM upshift, while the VAQ-2D film shows a Fermi level upshift and CBM downshift. Could the authors provide theoretical justification for the origins of the energy level shifts? Particularly, the CBM shift of the standard-2D films is very significant. What could cause this change?

[5] The discussions on the TRPL results need to be cautious and include all possibilities. For instance, the increased fast decay PL lifetime could be due to the inhibition of charge extraction to the electrodes as well as surface defect passivation. The slow decay PL lifetime may not necessarily be attributed to Shockley–Read–Hall (SRH) recombination.

[6] Could the authors perform a chemical composition analysis (e.g., XPS or TOF-SIMS) for the pristine and aged samples shown in Fig. 3? More evidence is needed to justify the phase segregation on the top surface and the impact of the 2D/3D interface on suppressing the surface halide segregation. Also, what are the clusters formed in the stand-2D perovskite sample after aging (Fig. 3j)? Could the authors characterize these clusters?

[7] The claim, "We also noted that the films deposited on HTL showed much more severe degradation than full devices. This further suggests that a balanced charge extraction improves the photostability of WBG perovskite solar cells" does not have supporting evidence. Could the authors provide data to support this statement?

[8] For the tandem MPP measurement, could the authors provide the actual temperature of the tandem cell under 1-sun illumination?

[9] In the Method sections, the vendor information for many materials is missing, such as DMAI, MAPbCl₃, GuaSCN, 2PACz, MeO-2PACz, and ITO NCs.

Reviewer #2 (Remarks to the Author):

In this manuscript, authors have developed a generic 3D-to-2D perovskite conversion approach via depositing a well-defined MAPbI₃ thin layer through a vapor-assisted two-step process, followed by its conversion into a 2D structure. Authors demonstrated that such a 2D/3D heterostructure could effectively suppress light-induced halide segregation, and reduce non-radiative interfacial recombination, and facilitate charge extraction. As a result, the wide-bandgap PSCs demonstrate a champion power conversion efficiency (PCE) of 19.6% and all-perovskite tandem solar cells exhibit a stabilized PCE of 28.1% and retain 90% of the initial performance after 855 hours of continuous 1-sun illumination. To me, this work is of great importance and may attract broad interest in the related fields. Thus, I strongly recommend the publication of this manuscript after minor revisions:

1. In Supplementary Fig. 2, why the XRD signal ($2\theta = 14^\circ$) split into doublet peaks after 500 h aging under OC conditions, and signals at other location remained unchanged compared to the fresh sample?
2. I am curious about the impact of VAQ-2D on the crystallization properties (i.e. crystallinity and grain size) of films. That would be more clear if the authors could provide XRD results.
3. I am also curious about the distribution of PEA in perovskite films. Will PEA cations diffuse into the film bulk?
4. Figure 4e compares the stability of the VAQ-2D devices with the control one, however, it would be desirable to add a comparison with the standard-2D treatment.
5. The design of the tandem solar cell is given in Figure 5a, but it is not clear how big the active area is. Please state the active area size.
6. Reporting data with areas larger than 1cm² has become a common consensus. Can the authors provide tandem cell data for 1cm² active area?

7. In Supplementary Fig. 6, could authors explain why a rise was observed in PL lifetime after light soaking when laser light was incident from the HTL side? Is it possible that trap states at HTL/perovskite interfaces were efficiently passivated via light soaking?
8. Could authors give more explanation on the mechanism why such vapor-assisted two-step process can realize a preferential growth of wider dimensionality ($n \geq 2$) atop wide-bandgap perovskite layers while standard methods couldn't?
9. In Fig. 3b-c, the TRPL spectrum of VAQ-2D/C60 exhibited rapider decay than control/C60 and standard-2D/C60 as a result of the improved charge extraction. But why the PLQY and PL intensity of VAQ-2D/C60 seemed to be nearly unchanged compared with the VAQ-2D film (Supplementary Fig. 15 and 16)? The PLQY and PL emission intensity are expected to be significantly reduced given the improved charge extraction.
10. In Fig. 4f, why the PCE of control device showed dramatic degradation in the first 50 h, and remained relatively stable during the subsequent 450 h?

Reviewer #3 (Remarks to the Author):

This manuscript presents a notable achievement: an efficient and photostable wide-bandgap (WBG) perovskite solar cell (PSC) designed for all-perovskite tandem solar cells, employing an innovative 3D-to-2D perovskite conversion approach. Impressively, a 28%-efficient all-perovskite tandem solar cell was also demonstrated accompanied by improved photostability and thermal stability. This work shows a very meaningful result in this field in that it not only shows excellent photovoltaic performance on one of the best all-perovskite tandem but also presents the 3D-to-2D converted heterojunction as a way, unlike conventional post-treatment approaches with defect passivation. In addition, this paper presents comprehensive experimental results for the photostability issue of WBG PSCs to well support the assertion as a whole. Therefore, I recommend that this paper would be published in Nature Communications after addressing the following concerns.

1. The 3D-to-2D transformation method involves multiple steps. Authors should provide more detailed PV performance after each individual step to aid readers in better understanding the contributions of individual steps to the final results and enhance the clarity of the findings.
2. Thickness of the PbI₂. It is not clear to me the reasoning behind the choice of 10 nm for device implementation. Additionally, how these thicknesses of PbI₂ are measured should be mentioned in the manuscript.

3. The authors mentioned the deposition of MAPbI₃ is crucial for the formation of quasi-2D. Authors should present the device performance without the MAI-solution step to better substantiate their proposed concept.

4. I am wondering if the PbI₂ which appeared after IPA washing (Supplementary Fig. 10) can be detected by X-ray diffraction. I recommend comparing XRD patterns both for the WBG perovskite and MAPbI₃ perovskite after IPA treatment to validate this observation.

Minor comments:

1. Some figures are described but not mentioned in the manuscript:

Fig. 2b, Fig. 2f, Fig. 4e, Fig. 5a

2. Some "Figure" in the manuscript should be written as "Fig."

3. In the method section, the author should check whether the volume ratio of DMF and DMSO in the NBG precursor solution is 2:1 or 3:1.

Point-by-point list of author actions in response to Reviewer comments

Manuscript #: NCOMMS-23-32660-T

We sincerely thank the reviewers for their much-valued suggestions, which have enabled us to improve the manuscript. Following are the detailed actions taken in light of reviewers' comments:

Reviewer #1 (Remarks to the Author):

In this research article, Wen et al. report a study of a heterojunction formed on 1.78-eV wide-bandgap perovskite via vapor-assisted 3D-to-2D perovskite conversion. This conversion involves depositing a well-defined methylammonium lead iodide (MAPbI₃) thin layer through a vapor-assisted two-step process and converting it into a 2D structure by reacting with phenethylammonium iodide (PEAI). This 2D/3D perovskite heterostructured interface suppresses light-induced halide segregation, reduces non-radiative interfacial recombination, and facilitates charge extraction, enabling efficient and stable single-junction wide-bandgap perovskite solar cells with a power conversion efficiency (PCE) of 19.6% and a high open-circuit voltage of 1.32 V. Using the photostable wide-bandgap perovskite subcell, the authors demonstrate all-perovskite tandem solar cells with an impressive stabilized PCE of 28.1%. The tandem cell retains 90% of its initial efficiency after 855 hours of continuous operation under 1-sun illumination.

This work is overall a well-organized study with a relatively novel method to introduce a 2D/3D heterojunction interface to stabilize wide-bandgap perovskites, despite the similarity in the approach to a recent publication from the same group (<https://www.nature.com/articles/s41586-023-06278-z>) and another work on all-inorganic perovskite (<https://www.nature.com/articles/s41560-022-01154-y>). Nonetheless, the work provides somewhat new insight into suppressing phase segregation in wide-bandgap perovskite and demonstrates stable and efficient all-perovskite tandem solar cells. The manuscript could be considered for publication if some technical issues can be adequately addressed.

Response: We sincerely thank the reviewer for a constructive peer review process.

[1] The section on photostability investigation of wide-bandgap perovskites needs to be clarified. One major concern is the lack of correlation and consistency between the 40 hours of 1-sun device light soaking experiment (Fig. 1c & d) and PL under 15-sun excitation for 10 min (Fig. 1e & f). The authors attempted to attribute the decreased J_{sc} under the Voc-aging condition (Fig. 1c) to light-induced phase segregation (Fig. 1e). However, the validation and interpretation of this hypothesis have two critical issues.

First, the aging conditions (1-sun for 40 h vs. 15-sun for 10 min) in Fig. 1 differ in power density and accumulated light fluence. The high power intensity of 15-sun is known to cause more severe degradation than 1-sun. This aging condition inconsistency also exists in supplementary experiments, i.e., 80, 500, and ~100 h for Supplementary Figures 1, 2, and 3, respectively. The design of the experiment needs better consistency in the aging time and conditions.

Response: In light of reviewer's suggestion, we now provide PL spectra of the WBG perovskite device under 1-Sun illumination within 1 hour in the revised manuscript:

“We used a 532 nm laser with an intensity equivalent to 15 suns, in order to accelerate the degradation process.”

“When the laser intensity is set to 1 sun, the difference in WBG device stability between OC and SC condition can still be observed.”

Supplementary Figure 3. PL spectra of perovskite device under illumination at (a) open-circuit and (b) short-circuit conditions. The samples were excited under a 532 nm laser for 60 min.

We have now standardized the aging time of 80 hours for the devices in Supplementary Figures 1, 2, and 3:

Supplementary Figure 2. XRD patterns of $\text{Cs}_{0.2}\text{FA}_{0.8}\text{Pb}(\text{I}_{0.6}\text{Br}_{0.4})_3$ perovskite devices before and after 80 h aging under MPP and OC conditions, respectively.

Supplementary Figure 3. Evolution of (a) V_{oc} , (b) J_{sc} , (c) FF, and (d) PCE of WBG perovskite solar cells under simulated one sun illumination with FACs/FAMA compositions. EQE evolution of (e) $\text{Cs}_{0.2}\text{FA}_{0.8}\text{Pb}(\text{I}_{0.6}\text{Br}_{0.4})_3$ and (f) $\text{MA}_{0.3}\text{FA}_{0.7}\text{Pb}(\text{I}_{0.6}\text{Br}_{0.4})_3$ devices under simulated one sun illumination.

Moreover, Fig. 1 indicates that degraded PV performance, particularly J_{sc} , is correlated with halide segregation, as evidenced in the PL peak shift. However, Supplementary Figure 4 shows no obvious changes in the PL spectrum of the cell after light soaking, even though the J_{sc} is reduced. Even if the phase segregation occurs and narrower bandgap I-rich phase forms, should the EQE cutoff shifts toward a longer wavelength corresponding to the I-rich phase? This is inconsistent with the hypothesis and needs a better explanation.

Response: We thank the reviewer for pointing out this. We acknowledge that there was an error in the testing presented in Figure S4, and we have retested the PL of the devices before and after aging 80 hours. We now correct the discussion in the revised manuscript:

"The intense emission peak originating from the enriched I-phase offers a credible explanation for the maintained V_{oc} ."

Supplementary Figure 4. PL spectra of $\text{Cs}_{0.2}\text{FA}_{0.8}\text{Pb}(\text{I}_{0.6}\text{Br}_{0.4})_3$ perovskite device before and after light soaking.

We applied a logarithmic transformation to the EQE curves of the devices before and after aging, revealing a pronounced emission peak attributed to the iodine-rich phase:

Figure R1. EQE curves of $\text{Cs}_{0.2}\text{FA}_{0.8}\text{Pb}(\text{I}_{0.6}\text{Br}_{0.4})_3$ perovskite device before and after light soaking.

[2] The statement "After light soaking, we noticed a rise in PL lifetime when laser light was incident from the HTL side, but there was barely any change in the PL decay when the laser was incident from the ETL side (Supplementary Fig. 6). The results suggest an inadequate transport of photogenerated electrons from the perovskite layer to ETL." needs a further explanation.

The data seems to indicate that light soaking does not significantly affect the perovskite/ETL interface (similar PL lifetimes before and after aging) but inhibits hole extraction at the perovskite/HTL interface (longer PL lifetime after aging), which conflicts with the claim.

Response: The penetration depth of the 375 nm laser was estimated to be <30 nm due to the high absorption coefficient of perovskites [J. Phys. Chem. Lett. 5, 1035–1039 (2014)]. Laser excitation from the ETL side predominantly generated excess carriers near the perovskite/ETL interface, with most photogenerated electrons rapidly extracted into the ETL. The corresponding TRPL decay reflected the transport of photogenerated holes from the perovskite to the HTL. Likewise, when laser excitation originated from the HTL side, the TRPL signal indicated the efficiency of electron transport to the ETL. Notably, under HTL-side laser excitation, we observed a pronounced increase in lifetime for the device after light soaking. This observation suggests that after degradation, it was the transport of electrons rather than holes that were impeded in the device [Nat Energy 6, 633–641 (2021)].

We now include a more specific explanation in the revised manuscript:

“Given the swift transfer of electrons (holes) generated at the interface towards adjacent transport layers, TRPL reveals the efficiency of holes (electrons) extraction to the opposite transport layer.”

[3] There are also some inconsistencies in the experimental design for the 2D/3D heterojunction section. The concentration of PEAI (1, 2, 5, 10 mg/ml) and evaporated PbI₂ thickness (10 and 20 nm) vary in different characterizations and are confusing.

The samples for XRD measurement (Fig. 2a) used 5 mg/ml PEAI with unknown PbI₂ thickness.

Response: We thank the reviewer for clarifying this. We acknowledge that our previous initial description may have caused confusion among readers. In both **Fig. 2a** and **Supplementary Figure 9**, the samples did not undergo PbI₂ vapor deposition. Instead, these experiments were conducted to directly investigate the formation of two-dimensional structures on the surfaces of different components following PEAI treatment. The use of PEAI solutions at various concentrations in **Supplementary Figure 9** was necessitated by the challenge of detecting 2D structures formed at lower concentrations (1 mg mL⁻¹) through XRD testing. Higher concentration solutions, such as 5 and 10 mg mL⁻¹, clearly illustrate the formation of exclusively n=1 2D structures on the WBG perovskite surface. Hence, we also employed a concentration of 5 mg mL⁻¹ in the experiments presented in **Fig. 2a**.

Through this experiment, we discovered that the surface of MAPbI₃ tends to produce 2D structures with $n \geq 2$, which benefits charge extraction at the perovskite/C₆₀ interface. However, when it comes to the surfaces of wide-bandgap (WBG) components containing Cs and Br, PEAI treatment exclusively yields a wider bandgap n=1 2D structure, which hampers electron transfer. These experiments underscore the substantial influence of PbI₂ and MAI on the formation of high n-value 2D structures. Consequently, we conceived the concept of transforming the subsequent thin layer of MAPbI₃ from a 3D to a 2D configuration atop WBG perovskite, as depicted in Figure 2c.

We now present a more accurate description of this process in the revised manuscript:

“We further explored the formation of 2D structures using PEAI solution spin-coated directly upon varying perovskite compositions, especially WBG compositions.”

“The application of PEAI solutions at various concentrations aimed to overcome the detection limitations for 2D structures formed at lower concentrations (1 mg mL⁻¹) by XRD testing, while higher concentrations (5 and 10 mg mL⁻¹) demonstrated the exclusive formation of n = 1 2D structures on the WBG perovskite surface.”

*“The XRD pattern confirmed the appearance of PbI₂ peaks in the MAPbI₃ film after IPA cleaning, while the Cs-Br WBG film remained unchanged (**Supplementary Fig. 12**).”*

“These experiments highlight the significant impact of PbI₂ and MAI on the formation of high n-value 2D structures, a phenomenon consistent with findings reported in the literature on 2D perovskites^{29,47,48}. ”

Later in the text, a condition of 10 nm PbI₂ and 2 mg/mL PEAI was used for synthesizing a 2D layer. The PL measurement then used 20 nm PbI₂ with an unknown PEAI concentration. Is the 20-nm PbI₂ film with 2D perovskite features shown in Fig. 2e used for the champion solar cell?

Could the authors specify how they determine the best PbI₂/PEAI combination for their device optimization and provide relevant characterization evidence?

Response: We thank the reviewer for pointing out this. We have now included an exploration of the process leading to the optimal device performance and the corresponding characterization analysis in **Supplementary Note 1**:

*“To achieve a better effect of improving the performance of the WBG PSC with a quasi-2D layer, we have optimized the processing of the VAQ-2D layer. We observed that PbI₂ layers thinner than 10 nm fail to form a continuous film on the surface of the 3D perovskite. To investigate the relationship between MAI concentration and the conversion of PbI₂ thickness to MAPbI₃, we conducted depositions of 10 nm and 20 nm PbI₂ on a glass substrate, as illustrated in **Supplementary Fig. 15**. Our findings revealed that MAI concentrations exceeding 2 mg mL⁻¹ can entirely convert 10 nm thick PbI₂, while concentrations exceeding 4 mg mL⁻¹ are necessary for full conversion of 20 nm thick PbI₂. Corresponding SEM images in **Supplementary Fig. 16** depict the transformation of 2 mg mL⁻¹ MAI with 10 nm thick PbI₂ into well-defined crystalline grains, whereas higher concentrations result in the formation of amorphous thin layers.*

*Given the inherent challenges in accurately characterizing thin 2D layers, we determined the optimal PEAI concentration based on device performance. As demonstrated in **Supplementary Figure 17-18**, the peak device performance is achieved with a combination of 10 nm thick PbI₂, 2 mg mL⁻¹ MAI, and 2 mg mL⁻¹ PEAI. Nevertheless, when the PbI₂ thickness is increased to 20 nm, a discernible reduction in both J_{sc} and FF is observed. This decline is attributed to the excessive thickness of the 2D layer, which hampers charge transport at the interface. Thus, we selected the combination of 10 nm thick PbI₂, 2 mg mL⁻¹ MAI, and 2 mg mL⁻¹ PEAI for the champion device.*

*While the diffraction peaks of PbI₂ and MAPbI₃ were detected during the optimized processing, the XRD pattern and PL emission from the resulting ultra-thin 2D perovskite layer were not observed (**Supplementary Fig. 19**). However, when the same processing was applied to deposit the 2D layer on glass, the XRD peak of VAQ-2D was detected, confirming the formation of 2D perovskite layers with n ≥ 2 (**Supplementary Fig. 19**). Additionally, when the PbI₂ layer was thickened to 20 nm, PL emission peaks at 560, 610, and 650 nm (corresponding to n = 2, 3, and 4, respectively) were observed in the films, providing further confirmation of the VAQ-2D structure.”*

Supplementary Figure 14. Top-view SEM image of WBG perovskite films with various PbI_2 thicknesses. We mark the thickness of the PbI_2 from the evaporation equipment (a) 1 nm (b) 3 nm (c) 5 nm and (d) 10 nm.

Supplementary Figure 15. XRD patterns of (a) 10 nm PbI_2 film and (b) 20 nm PbI_2 film deposited on the glass substrate and converted by MAI solution with various concentrations.

Supplementary Figure 16. Top-view SEM image of PbI₂ films deposited upon WBG perovskite film and reacting with various contents of MAI in the organic salt. **(a)** PbI₂ (10 nm) /MAI (2 mg mL⁻¹), **(b)** PbI₂ (10 nm) /MAI (3 mg mL⁻¹), **(c)** PbI₂ (20 nm) /MAI (4 mg mL⁻¹) and **(d)** PbI₂ (20 nm) /MAI (6 mg mL⁻¹).

Supplementary Figure 17. PV parameters of devices with different combinations of PbI₂, MAI and PEAI.

Supplementary Figure 18. EQE curves of WBG devices treated with VAQ-2D of varying thicknesses.

[4] It is interesting that the standard-2D layer induces a Fermi level downshift and CBM upshift, while the VAQ-2D film shows a Fermi level upshift and CBM downshift. Could the authors provide theoretical justification for the origins of the energy level shifts? Particularly, the CBM shift of the standard-2D films is very significant. What could cause this change?

Response: We appreciate the reviewer's clarification. The 2D perovskite structure generated with the PEA1 ligand is known as $\text{PEA}_2\text{MA}_{n-1}\text{Pb}_n\text{I}_{3n+1}$. Similar to 3D perovskites, the electronic properties of 2D perovskites are primarily determined by the inorganic component, governed by the $[\text{BX}_6]$ octahedral framework. VBM is made of an anti-bonding hybridization between Pb (6s) and I (5p) while CBM mainly consists of a bonding hybridization between Pb (6p) states [Chem 5, 10, 2593-2604 (2019)].

By adjusting the thickness of the inorganic layers (n-value) and B-X bond, the band structure in 2D perovskites can be effectively controlled. Specifically, reducing the thickness of the inorganic layers from a larger n-value to a smaller n-value results in an increasing bandgap due to size-induced quantum confinement [J. Am. Chem. Soc, 140, 2890–2896 (2018); Nano Lett, 17, 4759–4767 (2017)].

In the standard-2D with $n = 1$, the 2D structure exhibits the widest bandgap, approximately 2.4 eV, resulting in a significantly enhanced CBM. For the VAQ-2D structure, it has been confirmed for $n = 2, 3$, and 4. In this case, the bandgap of the 2D structure is smaller, causing the CBM to shift downward. Our observations are consistent with many previous reports on 2D perovskites [Science, 376,73–77 (2022); J. Am. Chem. Soc, 137, 7843–7850 (2015); Nat. Photon, 16, 352–358 (2022)].

[5] The discussions on the TRPL results need to be cautious and include all

possibilities. For instance, the increased fast decay PL lifetime could be due to the inhibition of charge extraction to the electrodes as well as surface defect passivation. The slow decay PL lifetime may not necessarily be attributed to Shockley–Read–Hall (SRH) recombination.

Response: We appreciate the reviewer's clarification. We now provide a more detailed and comprehensive discussion:

“Time-resolved PL (TRPL) spectra also indicate an increased lifetime of aged devices (Supplementary Fig. 6a). It's worth noting that the increased lifetime could be attributed to both aging effects and improved material quality. To delve deeper, we performed calculations for the differential lifetime (Supplementary Fig. 6b) to distinguish between charge extraction and trap-assisted recombination. The initial interval at shorter times is primarily influenced by the transfer of carriers from the bulk into the transport layer, while the subsequent interval at longer delay times is dominated by interfacial recombination. These results strongly suggest a suppressed carrier extraction towards the electrodes following the aging process – in connection with the significantly decreased J_{sc} and FF.”

Supplementary Figure 6. (a) TRPL spectra of $\text{Cs}_{0.2}\text{FA}_{0.8}\text{Pb}(\text{I}_{0.6}\text{Br}_{0.4})_3$ perovskite device before and after light soaking. (b) Computed differential lifetimes obtained by taking the derivative from fits to the transients in (a).

[6] Could the authors perform a chemical composition analysis (e.g., XPS or TOF-SIMS) for the pristine and aged samples shown in Fig. 3? More evidence is needed to justify the phase segregation on the top surface and the impact of the 2D/3D interface on suppressing the surface halide segregation. Also, what are the clusters formed in the stand-2D perovskite sample after aging (Fig. 3j)? Could the authors characterize these clusters?

Response: In light of reviewer’s suggestion, we now provide a more in-depth and comprehensive composition analysis in the revised manuscript:

“The Energy-dispersive X-ray spectroscopy (EDS) mapping (Supplementary Fig. 29-

31) shows that the distribution of I and Br in all samples was uniform before soaking. However, after soaking for 500 hours, control and standard-2D films exhibited non-uniform I and Br distributions (**Supplementary Fig. 29 and 30**). We speculate that the white clusters observed at SEM grain boundaries correspond to regions enriched with iodine after halide segregation in EDS mapping. Remarkably, the VAQ-2D film displayed minimal change in I and Br distribution after the same aging period. X-ray Photoelectron Spectroscopy (XPS) analysis of Pb 4f spectra revealed that the binding energies of Pb 4f_{5/2} and Pb 4f_{7/2} in both the control and standard-2D samples shifted towards lower binding energies by approximately 0.4 eV (**Supplementary Fig. 32**). This shift is attributed to a weakened binding between Pb and halide ions, likely due to the increased lattice volume resulting from halide segregation. However, in the case of the VAQ-2D film, the binding energy of Pb exhibited a minor change, approximately 0.05 eV, consistent with the EDS analysis, further emphasizing its enhanced photostability."

Supplementary Figure 29. SEM and EDS (Energy-dispersive X-ray spectroscopy) mapping of control film (a-d) before and (e-h) after soaking 500 h.

Supplementary Figure 30. SEM and EDS mapping of standard-2D film (a-d) before and (e-h) after soaking 500 h.

Supplementary Figure 31. SEM and EDS mapping of VAQ-2D film (a-d) before and (e-h) after soaking 500 h.

Supplementary Figure 32. XPS spectra of Pb 4f for (a) control, (b) standard-2D and (c) VAQ-2D before and after aging.

[7] The claim, "We also noted that the films deposited on HTL showed much more severe degradation than full devices. This further suggests that a balanced charge extraction improves the photostability of WBG perovskite solar cells" does not have supporting evidence. Could the authors provide data to support this statement?

Response: We appreciate the reviewer's clarification. We realize that this statement was incorrect, as aging under open-circuit conditions cannot represent the extraction process of the transport layer. Therefore, we have removed this statement in the revised manuscript.

[8] For the tandem MPP measurement, could the authors provide the actual temperature of the tandem cell under 1-sun illumination?

Response: We thank the reviewer for clarifying this. We now add the information on the actual temperature of the tandem cell during MPP operation in the revised manuscript:

"During MPP operation, the device's temperature is maintained at around 31 °C with the assistance of the fan cooling (Supplementary Fig. 42)."

Supplementary Figure 42. The temperature of the tandem device under MPP operation, measured by an infrared camera.

[9] In the Method sections, the vendor information for many materials is missing, such as DMAI, MAPbCl₃, GuaSCN, 2PACz, MeO-2PACz, and ITO NCs.

Response: We thank the reviewer for pointing out this. We now add the missing information in the Methods:

“All materials were used as received without further purification. PbI₂ (99.99%), PbBr₂ (99.99%), PbCl₂ (99.99%), 4PACz (>98.0%) and MeO-2PACz (>98.0%) were purchased from TCI Chemicals.”

“...and ITO NCs (<100 nm particle size (DLS) were purchased from Sigma-Aldrich.”

“The organic halide salts (FAI, MAI, PEAI, DMAI, MAcl) were purchased from GreatCell Solar Materials (Australia). Guanidinium thiocyanate (99%) and BCP (>99% sublimed) were purchased from Xi’an Polymer Light Technology Corp.”

Reviewer #2 (Remarks to the Author):

In this manuscript, authors have developed a generic 3D-to-2D perovskite conversion approach via depositing a well-defined MAPbI₃ thin layer through a vapor-assisted two-step process, followed by its conversion into a 2D structure. Authors demonstrated that such a 2D/3D heterostructure could effectively suppress light-induced halide segregation, and reduce non-radiative interfacial recombination, and facilitate charge extraction. As a result, the wide-bandgap PSCs demonstrate a champion power conversion efficiency (PCE) of 19.6% and all-perovskite tandem solar cells exhibit a stabilized PCE of 28.1% and retain 90% of the initial performance after 855 hours of continuous 1-sun illumination. To me, this work is of great importance and may attract broad interest in the related fields. Thus, I strongly recommend the publication of this manuscript after minor revisions:

Response: We thank the reviewer for a constructive peer review process.

1. In Supplementary Fig. 2, why the XRD signal ($2\theta = 14^\circ$) split into doublet peaks after 500 h aging under OC conditions, and signals at other location remained unchanged compared to the fresh sample?

Response: We thank the reviewer for pointing out this. Firstly, we confirmed that this phenomenon is not accidental, as it was repeated consistently in multiple experiments.

Figure R2. XRD patterns of $\text{Cs}_{0.2}\text{FA}_{0.8}\text{Pb}(\text{I}_{0.6}\text{Br}_{0.4})_3$ perovskite devices after 500 h light soaking from different batches.

Various crystal facets exhibit distinct atomic arrangements and coordination, resulting in diverse atomic potential landscapes. Consequently, these facets display disparate electronic, physical, and chemical properties. Several studies have highlighted variations in the stability (moisture, oxygen, etc.) of different crystal facets within perovskite thin films [Science, 379, 173–178 (2023); ACS Energy Lett, 7, 3120–3128 (2022)].

We speculate that the growth conditions and substrate used during the fabrication of perovskite films can also introduce larger strain in (100) orientation, making them less stable compared to other crystal faces [Nat Commun 11, 6328 (2020)].

2. I am curious about the impact of VAQ-2D on the crystallization properties (i.e. crystallinity and grain size) of films. That would be more clear if the authors could provide XRD results.

Response: In the light of reviewer's suggestion, we compare the XRD pattern of the WBG film with and without VAQ-2D treatment. The analysis of the full width at half maximum (FWHM) indicates that this thin layer of 2D did not introduce significant changes in the crystallization and orientation of the WBG film.

Figure R3. XRD patterns of control and VAQ-2D perovskite films deposited on NiO_x/Sam substrate.

3. I am also curious about the distribution of PEA in perovskite films. Will PEA cations diffuse into the film bulk?

Response: We thank the reviewer for clarifying this. We now provide the SIMS analysis of WBG film with the VAQ-2D layer.

The time-of-flight secondary ion mass spectrometry (TOF-SIMS) depth profile of the VAQ-2D device revealed that PEA is mainly distributed on the top interface within 20 nm.

Figure R4. TOF-SIMS of WBG perovskite films with VAQ-2D layer deposited on NiO_x/ ITO substrate.

4. Figure 4e compares the stability of the VAQ-2D devices with the control one,

however, it would be desirable to add a comparison with the standard-2D treatment.

Response: We now include the operation stability of standard-2D device in the revised Supporting Information.

The following explanation was also included in the revised manuscript:

*“The stability of standard-D devices, as shown in **Supplementary Fig. 38**, deteriorated more rapidly compared to the control devices, and we attribute this to the degradation caused by interface electron blocking.”*

Supplementary Figure 38. MPP tracking was measured with the encapsulated control and standard-2D devices under full solar illumination (AM 1.5G, 100 mW cm⁻²) in ambient conditions.

5. The design of the tandem solar cell is given in Figure 5a, but it is not clear how big the active area is. Please state the active area size.

Response: In light of reviewer’s suggestion, we now state the active area size of the tandem device in the revised manuscript:

“We fabricated 30 tandem devices (area ~0.49 cm²), showing an average PCE of 27.7 ± 0.3% and a relatively narrow PCE distribution.”

6. Reporting data with areas larger than 1cm² has become a common consensus. Can the authors provide tandem cell data for 1cm² active area?

Response: In light of the reviewer’s suggestion, we fabricated large-area tandem devices (1.05 cm² aperture area). The PCE distribution and champion PCE were provided in **Supplementary Fig. 41**:

“We also achieved a champion PCE of 27.3% for large-area tandem (1.05 cm² aperture area), with an average PCE of 26.7 ± 0.3%.”

Supplementary Figure 41. (a) J - V curves of the champion tandem cell with an aperture area of 1.05 cm^2 . (b) PCE distribution of 30 tandem devices (1.05 cm^2), showing an average PCE of $26.7 \pm 0.3\%$.

7. In Supplementary Fig. 6, could authors explain why a rise was observed in PL lifetime after light soaking when laser light was incident from the HTL side? Is it possible that trap states at HTL/perovskite interfaces were efficiently passivated via light soaking?

Response: We thank the reviewer for clarifying this. Due to the sharp decline in device efficiency after soaking, we consider that defect trap states have increased.

In Supplementary Fig. 6, the penetration depth of the 375 nm laser was estimated to be $<30 \text{ nm}$ due to the high absorption coefficient of perovskites. Laser excitation from the ETL side predominantly generated excess carriers near the perovskite/ETL interface, with most photogenerated electrons rapidly extracted into the ETL. The corresponding TRPL decay reflected the transport of photogenerated holes from the perovskite to the HTL. Likewise, when laser excitation originated from the HTL side, the TRPL signal indicated the efficiency of electron transport to the ETL. Notably, under HTL-side laser excitation, we observed a pronounced increase in lifetime for the device after light soaking. This observation suggests that after degradation, it was the transport of electrons rather than holes that were impeded in the device. [Nat Energy 6, 633–641 (2021)]

We now include a more specific explanation in the revised manuscript:

“Given the swift transfer of electrons (holes) generated at the interface towards adjacent transport layers, TRPL reveals the efficiency of holes (electrons) extraction to the opposite transport layer.”

8. Could authors give more explanation on the mechanism why such vapor-assisted two-step process can realize a preferential growth of wider

dimensionality ($n \geq 2$) atop wide-bandgap perovskite layers while standard methods couldn't?

Response: We thank the reviewer for pointing out this. **Fig. R5** provides direct evidence of the crucial role of MAI in the formation of high n -value 2D structures. Since the surface of CsBr-riched WBG lacks MA^+ participating in the process of 2D restructuring, standard-2D method couldn't form high n -value 2D structures.

Figure R5. XRD patterns of 10 nm PbI_2 converted by PEAI solution with and without MAI step.

We now provide a more robust and detailed mechanistic explanation in the revised manuscript:

“The XRD pattern confirmed the appearance of PbI_2 peaks in the MAPbI_3 film after IPA cleaning, while the Cs-Br WBG film remained unchanged (Supplementary Fig. 12).”

“These experiments highlight the significant impact of PbI_2 and MAI on the formation of high n -value 2D structures, a phenomenon consistent with findings reported in the literature on 2D perovskites^{29,47,48}.”

Supplementary Figure 12. XRD pattern of (a) MAPbI₃, (b) Cs_{0.2}FA_{0.8}PbI_{1.8}Br_{1.2} film before and after IPA washing.

9. In Fig. 3b–c, the TRPL spectrum of VAQ–2D/C60 exhibited rapid decay than control/C60 and standard–2D/C60 as a result of the improved charge extraction. But why the PLQY and PL intensity of VAQ–2D/C60 seemed to be nearly unchanged compared with the VAQ–2D film (Supplementary Fig. 15 and 16)? The PLQY and PL emission intensity are expected to be significantly reduced given the improved charge extraction.

Response: We appreciate the reviewer's clarification. This question is frequently encountered and holds significant importance in our research. Let us consider a scenario in which an electron is extracted to the C₆₀ material during PL experiments. However, this alone would not extinguish the PL emission unless there exists an interface recombination process or recombination within the ETL. Given that the illumination is in a steady-state condition, an equal number of electrons would be re-injected into the perovskite. One plausible scenario involves the electron transferring to the C₆₀ material and subsequently recombining within it. While this would "quench" the PL emission, it remains a recombination event, which is detrimental to the V_{oc} .

Therefore, any decrease in PLQY or PL in these perovskite stacks should not be considered as evidence of charge extraction; it may simply indicate increased non-radiative recombination. [Adv. Energy Mater, 2201109 (2022); Adv. Energy Mater, 11, 2003489 (2021); Adv. Energy Mater, 10, 1904134 (2020)]

10. In Fig. 4f, why the PCE of control device showed dramatic degradation in the first 50 h, and remained relatively stable during the subsequent 450 h?

Response: In a recent publication [Nat Commun 14, 4869 (2023)], Antonio et al. used the unsupervised machine learning method self-organising map and summarized four types of degradation curve shapes. The degradation pattern exhibited by the control device in Fig. 4f conforms to fast-exponential decay. Our hypothesis suggests a potential correlation with the speed of halide segregation dynamics.

The degradation of perovskite devices is a complex process, and we look forward to the development of more advanced in-situ characterization and analysis techniques to better understand the underlying mechanisms.

Reviewer #3 (Remarks to the Author):

This manuscript presents a notable achievement: an efficient and photostable wide-bandgap (WBG) perovskite solar cell (PSC) designed for all-perovskite tandem solar cells, employing an innovative 3D-to-2D perovskite conversion approach. Impressively, a 28%-efficient all-perovskite tandem solar cell was also demonstrated accompanied by improved photostability and thermal stability. This work shows a very meaningful result in this field in that it not only shows excellent photovoltaic performance on one of the best all-perovskite tandem but also presents the 3D-to-2D converted heterojunction as a way, unlike conventional post-treatment approaches with defect passivation. In addition, this paper presents comprehensive experimental results for the photostability issue of WBG PSCs to well support the assertion as a whole. Therefore, I recommend that this paper would be published in Nature Communications after addressing the following concerns.

Response: We sincerely thank the reviewer for a constructive peer review process.

1. The 3D-to-2D transformation method involves multiple steps. Authors should provide more detailed PV performance after each individual step to aid readers in better understanding the contributions of individual steps to the final results and enhance the clarity of the findings.

Response: We thank the reviewer for clarifying this. We now add the detailed statistical photovoltaic performance of devices in **Supplementary Fig. 34** and provide the discussion in **Supplementary Note 2**:

“Supplementary Fig. 34 illustrates the evolution of device performance during the three-step fabrication of VAQ-2D. Following the deposition of PbI_2 , the device performance shows an increase in V_{oc} and FF. However, due to the wider energy levels impeding charge transport, the J_{sc} exhibits a significant decrease. After MAI treatment, the surface transitions to the narrower energy levels of $MAPbI_3$, resulting in an increase in J_{sc} but decreased V_{oc} and FF. Finally, after PEAI treatment, the surface achieves an optimal energy level, resulting in a substantial enhancement in all three device parameters.”

Supplementary Figure 34. Performance of WBG perovskite solar cells during vapor-assisted 3D-to-2D conversion method. Statistical performance of 15 devices for each type is presented. The optimal thickness of the evaporated PbI₂ layer is 10 nm, which is subsequently transformed into a MAPbI₃ perovskite layer after organic-salt (MAI) treatment. Finally, it is converted into a quasi-two-dimensional structure through PEAI treatment.

2. Thickness of the PbI₂. It is not clear to me the reasoning behind the choice of 10 nm for device implementation. Additionally, how these thicknesses of PbI₂ are measured should be mentioned in the manuscript.

Response: We appreciate the reviewer's clarification. We note that we reference the thickness of the PbI₂ layer as determined by the evaporation equipment. This is due to the challenges in accurately measuring the thickness of such a thin PbI₂ layer using SEM measurements.

To ensure accuracy, we calibrated the deposition rate in the evaporation equipment by measuring the thickness of the inorganic layer fabricated on a flat bare ITO glass substrate. We now better explain the measurement of thickness in the **Methods** section in the revised manuscript:

“The thickness of PbI₂ layer was read from the evaporation equipment (thickness was calibrated by the evaporation of films on a flat ITO glass substrate).”

When the thickness of the PbI_2 inorganic layer is less than 10 nm, it cannot effectively form a continuous and complete coverage atop the Pb-Sn perovskite layer, as illustrated in **Supplementary Fig. 14**:

Supplementary Figure 14. Top-view SEM image of WBG perovskite films with various PbI_2 thicknesses. We mark the thickness of the PbI_2 from the evaporation equipment (a) 1 nm (b) 3 nm (c) 5 nm and (d) 10 nm.

When the thickness exceeds 10 nm, we selected the optimal parameters based on the device's performance. We now add the statistical photovoltaic performance of devices with various PbI_2 thicknesses and provide a more detailed discussion in **Supplementary Note 1**:

*“As demonstrated in **Supplementary Figure 17-18**, the peak device performance is achieved with a combination of 10 nm thick PbI_2 , 2 mg mL⁻¹ MAI, and 2 mg mL⁻¹ PEAI. Nevertheless, when the PbI_2 thickness is increased to 20 nm, a discernible reduction in both J_{sc} and FF is observed. This decline is attributed to the excessive thickness of the 2D layer, which hampers charge transport at the interface. Thus, we selected the combination of 10 nm thick PbI_2 , 2 mg mL⁻¹ MAI, and 2 mg mL⁻¹ PEAI for the champion device.”*

Supplementary Figure 17. PV parameters of devices with different combinations of PbI₂, MAI and PEAI.

Supplementary Figure 18. EQE curves of WBG devices treated with VAQ-2D of varying thicknesses.

3. The authors mentioned the deposition of MAPbI₃ is crucial for the formation of quasi-2D. Authors should present the device performance without the MAI-solution step to better substantiate their proposed concept.

Response: In light of the reviewer’s suggestion, we compare the device performance with various 2D treatments in **Supplementary Note 2**:

“In the absence of the MAI step, devices with a deposited layer of PbI₂ followed by PEAI treatment displayed a more pronounced reduction in current compared to those

directly spin-coated with PEAI (**Supplementary Fig. 33**). This effect can be attributed to the thicker $n=1$ 2D layer impeding charge transport, emphasizing the essential role of MA^+ in the formation of higher n -value 2D structures.”

Supplementary Figure 33. Performance of WBG perovskite solar cells with various 2D treatments: PEAI solution, vaped-PbI₂ + PEAI solution, vaped-PbI₂ + PEAI solution, vaped-PbI₂ + MAI + PEAI solution.

4. I am wondering if the PbI₂ which appeared after IPA washing (Supplementary Fig. 10) can be detected by X-ray diffraction. I recommend comparing XRD patterns both for the WBG perovskite and MAPbI₃ perovskite after IPA treatment to validate this observation.

Response: We now include the XRD pattern of MAPbI₃ and Cs_{0.2}FA_{0.8}PbI_{1.8}Br_{1.2} film in the revised manuscript:

“The XRD pattern confirmed the appearance of PbI₂ peaks in the MAPbI₃ film after IPA cleaning, while the Cs-Br WBG film remained unchanged (**Supplementary Fig. 12**).”

Supplementary Figure 12. XRD pattern of (a) MAPbI₃, (b) Cs_{0.2}FA_{0.8}PbI_{1.8}Br_{1.2} film before and after IPA washing.

Minor comments:

1. Some figures are described but not mentioned in the manuscript:
Fig. 2b, Fig. 2f, Fig. 4e, Fig. 5a

Response: We have added the missing figures in the revised manuscript.

2. Some "Figure" in the manuscript should be written as "Fig."

Response: We have corrected the typos in the revised manuscript.

3. In the method section, the author should check whether the volume ratio of DMF and DMSO in the NBG precursor solution is 2:1 or 3:1.

Response: The volume ratio of DMF and DMSO in the NBG precursor solution is 2:1. We have deleted the inaccurate description in the revised manuscript.

REVIEWERS' COMMENTS

Reviewer #1 (Remarks to the Author):

The authors have adequately addressed my comments. I recommend this revised manuscript for publication as is.

Reviewer #2 (Remarks to the Author):

In this revised manuscript titled "Heterojunction formed via 3D-to-2D perovskite conversion for photostable wide-bandgap perovskite solar cells", the authors have addressed all my concerns and questions. The overall quality has improved to the point where it is ready for publication.

Reviewer #3 (Remarks to the Author):

The authors have carefully addressed most of my concerns and the manuscript can be published in its current form.